# Automatic Symmetry Discovery with Lie Algebra Convolutional Network

**Nima Dehmamy**
Northwestern University
nimadt@bu.edu

**Robin Walters**
Northeastern University
rwalters@northeastern.edu

**Yanchen Liu**
Northeastern University
liu.yanc@northeastern.edu

**Dashun Wang**
Northwestern University
dashun.wang@kellogg.northwestern.edu

**Rose Yu**
University of California San Diego
roseyu@ucsd.edu

## Abstract

Existing equivariant neural networks require prior knowledge of the symmetry group and discretization for continuous groups. We propose to work with Lie algebras (infinitesimal generators) instead of Lie groups. Our model, the Lie algebra convolutional network (L-conv) can automatically discover symmetries and does not require discretization of the group. We show that L-conv can serve as a building block to construct *any* group equivariant feedforward architecture. Both CNNs and Graph Convolutional Networks can be expressed as L-conv with appropriate groups. We discover direct connections between L-conv and physics: (1) group invariant loss generalizes field theory (2) Euler-Lagrange equation measures the robustness, and (3) equivariance leads to conservation laws and Noether current. These connections open up new avenues for designing more general equivariant networks and applying them to important problems in physical sciences.[1]

## 1 Introduction

Incorporating symmetries into a deep learning architecture can reduce sample complexity, improve generalization, while significantly decreasing the number of model parameters (Cohen et al., 2019b; Cohen & Welling, 2016b; Ravanbakhsh et al., 2017; Ravanbakhsh, 2020; Wang et al., 2020). For instance, Convolutional Neural Networks (CNN) (LeCun et al., 1989, 1998) implement translation symmetry through weight sharing. General principles for constructing symmetry-aware group equivariant neural networks were introduced in Cohen & Welling (2016b), Kondor & Trivedi (2018), and Cohen et al. (2019b).

However, most work on equivariant networks requires knowing the symmetry group *a priori*. A different equivariant model needs to be re-designed for each symmetry group. In practice, we may not have a good inductive bias and such knowledge of the symmetries may not be available. Constructing and selecting the equivariant network with the appropriate symmetry group becomes quite tedious. Furthermore, many existing works are limited to *finite groups* such as permutations Hartford et al. (2018); Ravanbakhsh et al. (2017); Zaheer et al. (2017), 90 degree rotations Cohen et al. (2018) or dihedral groups $D_N$ and $E(2)$ Weiler & Cesa (2019).

---

[1]Code: github.com/nimadehmamy/L-conv-code

For a continuous group, existing approaches either discretize the group Weiler et al. (2018a,b); Cohen & Welling (2016a), or use a truncated sum over irreducible representations (irreps) Weiler & Cesa (2019); Weiler et al. (2018a) via spherical harmonics in Worrall et al. (2017) or more general Clebsch-Gordon coefficients Kondor et al. (2018); Bogatskiy et al. (2020). These approaches are prone to approximation error. Recently, Finzi et al. (2020) propose to approximates the integral over the Lie group by Monte Carlo sampling. This approach requires implementing the matrix exponential and obtaining a local neighborhood for each point. Both parametrizing Lie groups for sampling and finding irreps are computationally expensive. Finzi et al. (2021) provide a general algorithm for constructing equivariant multi-layer perceptrons (MLP), but require explicit knowledge of the group to encode its irreps, and solving a set of constraints.

We provide a novel framework for designing equivariant neural networks. We leverage the fact that Lie groups can be constructed from a set of infinitesimal generators, called Lie algebras. A Lie algebra has a finite basis, assuming the group is finite-dimensional. Working with the Lie algebra basis allows us to encode an infinite group without discretizing or summing over irreps. Additionally, all Lie algebras have the same general structure and hence can be implemented the same way. We propose Lie Algebra Convolutional Network (**L-conv**), a novel architecture that can automatically discover symmetries from data. Our main contributions can be summarized as follows:

- We propose the Lie algebra convolutional network (**L-conv**), a building block for constructing group equivariant neural networks.
- We prove that multi-layer L-conv can approximate group convolutional layers, including CNNs, and find graph convolutional networks to be a special case of L-conv.
- We can learn the Lie algebra basis in L-conv, enabling automatic symmetry discovery.
- L-conv also reveals interesting connections between physics and learning: equivariant loss generalizes important Lagrangians in field theory; robustness and equivariance can be expressed as Euler-Lagrange equations and Noether currents.

Learning symmetries from data has been studied in limited settings for commutative Lie groups as in Cohen & Welling (2014), 2D rotations and translations in Rao & Ruderman (1999), Sohl-Dickstein et al. (2010) or permutations (Anselmi et al., 2019). In the non-commutative case, GeoManCEr (Pfau et al., 2020) uses data points related by small transformations to learn non-abelian Lie groups, but it does not introduce an equivariant layer architecture. (Zhou et al., 2020) propose a general method for symmetry discovery. Yet, their weight-sharing scheme and the symmetry generators are very different from ours. Our approach use much fewer parameters and has a direct interpretation using Lie algebras (SI B.3). Benton et al. (2020) propose Augerino to learn a distribution over data augmentations. It also involves Lie algebras, but is restricted to a subgroup of 2D affine transformations and requires matrix logarithm and sampling (SI B.3). In contrast, our approach is simpler and more general. Our approach uses composition of small transformations to achieve large transformations. In this sense bears some resemblance to symnets (Gens & Domingos, 2014), but the rest of the construction is different.

## 2 Background

We review the core concepts L-conv builds upon: equivariance, group convolution and Lie algebras.

**Notations.** Unless explicitly stated, $a$ in $A^a$ is an index, not an exponent. We use the Einstein summation $A^a B_{ab} = \sum_a A^a B_{ab} = [AB]_b$, where a repeated upper and lower index are summed.

**Equivariance.** Let $\mathcal{S}$ be a topological space on which a Lie group $G$ (continuous group) acts from the left, meaning for all $\boldsymbol{x} \in \mathcal{S}$ and $g \in G$, $g\boldsymbol{x} \in \mathcal{S}$. We refer to $\mathcal{S}$ as the base space. Let $\mathcal{F}$, the "feature space", be the vector space $\mathcal{F} = \mathbb{R}^m$. Each data point is a feature map $f : \mathcal{S} \to \mathcal{F}$. The action of $G$ on the input of $f$ induces an action on feature maps. For "scalar" features, for $u \in G$, the transformed features $u \cdot f$ are given by

$$u \cdot f(\boldsymbol{x}) = f(u^{-1}\boldsymbol{x}). \tag{1}$$

Denote the space of all functions from $\mathcal{S}$ to $\mathcal{F}$ by $\mathcal{F}^\mathcal{S}$, so that $f \in \mathcal{F}^\mathcal{S}$. Let $F$ be a mapping to a new feature space $\mathcal{F}' = \mathbb{R}^{m'}$, meaning $F : \mathcal{F}^\mathcal{S} \to \mathcal{F}'^\mathcal{S}$. We say $F$ is *equivariant* under $G$ if $G$ acts on $\mathcal{F}'$

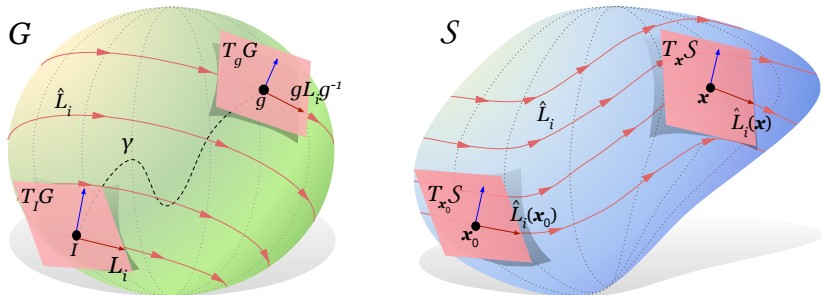

Figure 1: **Lie group and Lie algebra:** Illustration of the group manifold of a Lie group $G$ (left). The Lie algebra $\mathfrak{g} = T_I G$ is the tangent space at the identity $I$. $L_i$ are a basis for $T_I G$. If $G$ is connected, $\forall g \in G$ there exist paths like $\gamma$ from $I$ to $g$ and $g$ can be written as a path-ordered integral $g = P \exp[\int_\gamma dt^i L_i]$. **Base space** Right is a schematic of the base space $\mathcal{S}$ as a manifold. The lift $\boldsymbol{x} = g\boldsymbol{x}_0$ takes $\boldsymbol{x} \in \mathcal{S}$ to $g \in G$, and maps the tangent spaces $T_{\boldsymbol{x}}\mathcal{S} \to T_g G$. Each Lie algebra basis $L_i \in \mathfrak{g} = T_I G$ generates a vector field $\hat{L}_i$ on the tangent bundle $TG$ via the pushforward $\hat{L}_i(g) = g L_i g^{-1}$. Via the lift, $L_i$ also generates a vector field $\hat{L}_i = \hat{L}_i^\alpha(\boldsymbol{x})\partial_\alpha = [g L_i \boldsymbol{x}_0]^\alpha \partial_\alpha$.

and for $u \in G$, we have

$$u \cdot (F(f)) = F(u \cdot f). \tag{2}$$

**Group Convolution.** Kondor & Trivedi (2018) showed that $F$ is a linear equivariant map if and only if it performs a group convolution (G-conv). To define G-conv, we first lift $\boldsymbol{x}$ to elements in $G$ (Kondor & Trivedi, 2018). Specifically, we pick an origin $\boldsymbol{x}_0 \in \mathcal{S}$ and replace each point $\boldsymbol{x} = g\boldsymbol{x}_0$ by $g$. We will often drop $\boldsymbol{x}_0$ for brevity and write $f(g) \equiv f(g\boldsymbol{x}_0)$. Let $\kappa : G \to \mathbb{R}^{m'} \otimes \mathbb{R}^m$ be a linear transformation from $\mathcal{F}$ to $\mathcal{F}'$. G-conv is defined as

$$[\kappa \star f](g) = \int_G \kappa(g^{-1}v)f(v)dv = \int_G \kappa(v)f(gv)dv, \tag{3}$$

We denote the Haar measure on $G$ as $dv \equiv d\mu(v)$ for brevity.

**Equivariance of G-conv.** G-conv in equation 3 is equivariant (Kondor & Trivedi, 2018). By definition, for $w \in G$ we have

$$[\kappa \star w \cdot f](g) = \int_G \kappa(v)w \cdot f(gv)dv = \int_G \kappa(v)f(w^{-1}gv)dv$$
$$= [\kappa \star f](w^{-1}g) = w \cdot [\kappa \star f](g) \tag{4}$$

Existing works on equivariance networks implement $\int_G$ by discretizing the group or summing over irreps. We take a different approach and use the infinitesimal generators of the group. While a Lie group $G$ is infinite, usually it can be generated using a small number of infinitesimal generator, comprising its "Lie algebra". We use the Lie algebra to introduce a building block to approximate G-conv. Figure 1 visualizes a Lie group, Lie algebra and the concept we discuss below.

**Lie algebra.** Let $G$ be a Lie group, which includes common continuous groups. Group elements $u \in G$ infinitesimally close to the identity element $I$ can be written as $u \approx I + \epsilon^i L_i$ (note Einstein summation), where $L_i \in \mathfrak{g}$ with the Lie algebra $\mathfrak{g} = T_I G$ is the tangent space of $G$ at the identity element. The Lie algebra has the property that it is closed under a Lie bracket $[\cdot, \cdot] : \mathfrak{g} \times \mathfrak{g} \to \mathfrak{g}$

$$[L_i, L_j] = c_{ij}{}^k L_k, \tag{5}$$

which is skew-symmetric and satisfies the Jacobi identity. Here the coefficients $c_{ij}{}^k \in \mathbb{R}$ or $\mathbb{C}$ are called the structure constants of the Lie algebra. For matrix representations of $\mathfrak{g}$, $[L_i, L_j] = L_i L_j - L_j L_i$ is the commutator. The $L_i$ are called the infinitesimal generators of the Lie group.

**Exponential map.** If the manifold of $G$ is connected [2], an exponential map $\exp : \mathfrak{g} \to G$ can be defined such that $g = \exp[t^i L_i] \in G$. For matrix groups, if $G$ is connected and compact, the matrix

---

[2] When $G$ has multiple connected components, these results hold for the component containing $I$, and generalize easily for multi-component groups such as $\mathbb{Z}_k \otimes G$ (Finzi et al., 2021).

exponential is such a map and it is surjective. For most other groups (except $\text{GL}_d(\mathbb{C})$ and nilpotent groups) it is not surjective. Nevertheless, for any connected group every $g \in G$ can be written as a product $g = \prod_a \exp[t_a^i L_i]$ (Hall, 2015). Making $t_a^i$ infinitesimal steps $dt^i(s)$ tangent to a path $\gamma$ from $I$ to $g$ on $G$ yields the surjective path-ordered exponential in physics, denoted as $g = P \exp[\int_\gamma dt^i L_i]$ (SI A, and see Time-ordering in Weinberg (1995, p143)).

**Pushforward.** $L_i \in T_I G$ can be pushed forward to $\hat{L}_i(g) = g L_i g^{-1} \in T_g G$ to form a basis for $T_g G$, satisfying the same Lie algebra $[\hat{L}_i(g), \hat{L}_j(g)] = c_{ij}{}^k \hat{L}_k(g)$. The manifold of $G$ together with the set of all $T_g G$ attached to each $g$ forms the tangent bundle $TG$, a type of fiber bundle (Lee et al., 2009). $\hat{L}_i$ is a vector field on $TG$. The lift maps $\hat{L}_i$ to an equivalent vector field on $T\mathcal{S}$, which we will also denote by $\hat{L}_i$. Figure 1 illustrates the flow of these vector fields on $TG$ and $T\mathcal{S}$.

## 3    Lie Algebra Convolutional Network

We can use the Lie algebra basis $L_i \in \mathfrak{g}$ to construct the Lie group $G$ with the exponential map. Similarly, we show that Lie algebras can also serve as building blocks to construct G-conv layers. We propose the Lie algebra convolutional network (L-conv). The key idea is to approximate the kernel $\kappa(u)$ using localized kernels which can be constructed using the Lie algebra (Fig. 2). This is possible because the exponential map is a generalization of a Taylor expansion. We show that a G-conv whose kernel is concentrated near the identity can be expanded in the Lie algebra.

Let $\delta_\eta(u) \in \mathbb{R}$ denote a normalized *localized kernel*, meaning $\int_G \delta_\eta(g) dg = 1$, and with support on a small neighborhood of size $\eta$ centered around the identity $I$ (i.e., $\delta_\eta(I + \epsilon^i L_i) \to 0$ if $\|\epsilon\|^2 > \eta^2$). We pick $\delta_\eta(v_\epsilon) \sim \theta(\eta^2 - \|\epsilon\|^2)$, for $v_\epsilon = I + \epsilon^i L_i \in T_I G$ and $\delta_\eta(v) = 0$ for all other $v \notin T_I G$ ($\theta(\cdot)$ being the Heaviside step function). Let $\kappa_0 : G \to \mathbb{R}^{m'} \otimes \mathbb{R}^m$ be given by

$$[\kappa_0]_a^b (u) = \left[W^0\right]_a^c \delta_\eta \left(u \left(I - \left[\bar{\epsilon}^i\right]_c^b L_i\right)\right) \tag{6}$$

where $W^0 \in \mathbb{R}^{m'} \otimes \mathbb{R}^h$ and $\bar{\epsilon}^i \in \mathbb{R}^h \otimes \mathbb{R}^m$ are constants, and we choose $\left|[\bar{\epsilon}^i]_b^a\right| < \eta$. Note that $(I + \epsilon^i L_i)(I - \bar{\epsilon}^j L_j) = I + [\epsilon - \bar{\epsilon}]^i L_i + O(\eta^2)$. Therefore,

$$\int \epsilon^i d\epsilon \delta_\eta \left((I + \epsilon^i L_i)(I - \bar{\epsilon}^j L_j)\right) = \bar{\epsilon}^i \tag{7}$$

The localized kernels $\kappa_0$ can be used to approximate G-conv.

**Linear expansion of G-conv with localized kernel.** We can expand a G-conv whose kernel is $\kappa_0(u) = W^0 \delta_\eta(u)$ in the Lie algebra of $G$ to linear order. With $v_\epsilon = I + \epsilon^i L_i$, we have (see SI A)

$$Q[f](g) = [\kappa_0 \star f](g) = \int_G dv \kappa_0(v) f(gv) = \int_{\|\epsilon\|<\eta} dv_\epsilon \kappa_0(v_\epsilon) f(gv_\epsilon)$$

$$= W^0 \int d\epsilon \delta_\eta(v_\epsilon) \left[f(g) + \epsilon^i g L_i \cdot \frac{d}{dg} f(g) + O(\epsilon^2)\right]$$

$$= W^0 \left[I + \bar{\epsilon}^i g L_i \cdot \frac{d}{dg}\right] f(g) + O(\eta^2) \tag{8}$$

with $W^0 \in \mathbb{R}^{m'} \otimes \mathbb{R}^h$ and $\bar{\epsilon}^i \in \mathbb{R}^h \otimes \mathbb{R}^m$, as before. Here $d\epsilon$ is the integration measure on the Lie algebra $\mathfrak{g} = T_I G$ induced by the Haar measure $dv_\epsilon$ on $G$.

**Interpreting the derivatives.** In a matrix representation of $G$, we have $g L_i \cdot \frac{df}{dg} = [g L_i]_\alpha^\beta \frac{df}{dg_\alpha^\beta} = \text{Tr}\left[[g L_i]^T \frac{df}{dg}\right]$. This can be written in terms of partial derivatives $\partial_\alpha f(\boldsymbol{x}) = \partial f / \partial x^\alpha$ as follows. Using $\boldsymbol{x}^\rho = g_\sigma^\rho \boldsymbol{x}_0^\sigma$, we have $\frac{df(g\boldsymbol{x}_0)}{dg_\beta^\alpha} = \boldsymbol{x}_0^\beta \partial_\alpha f(\boldsymbol{x})$, and so

$$\hat{L}_i f(\boldsymbol{x}) \equiv g L_i \cdot \frac{df}{dg} = [g L_i]_\beta^\alpha \boldsymbol{x}_0^\beta \partial_\alpha f(\boldsymbol{x}) = [g L_i \boldsymbol{x}_0] \cdot \nabla f(\boldsymbol{x}) \tag{9}$$

Hence, for each $L_i$, the pushforward $g L_i g^{-1}$ generates a flow on $\mathcal{S}$ through the vector field $\hat{L}_i \equiv g L_i \cdot d/dg = [g L_i g^{-1} \boldsymbol{x}]^\alpha \partial_\alpha$ (Fig. 1).

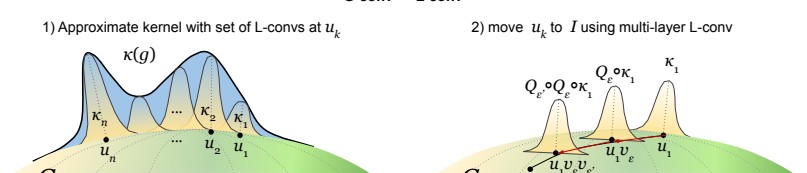

Figure 2: Sketch of the procedure for approximating G-conv using L-conv. First, the kernel is written as the sum of a number of localized kernels $\kappa_k$ with support around $u_k$ (left). Each of the $\kappa_k$ is then moved toward identity by composing multiple L-conv layers $Q_{\epsilon'} \circ Q_\epsilon \ldots \kappa_k$ (right).

**Lie algebra convolutional (L-conv) layer.** Equation 8 states that for a kernel localized near the identity, the effect of the kernel can be summarized in $W^0$ and $\bar{\epsilon}^i \hat{L}_i$. Note that we do not need to perform the integral over $G$ explicitly anymore. Instead of working with a kernel $\kappa_0$, we only need to specify $W^0$ and $\bar{\epsilon}^i$. Hence, in general, we define the Lie algebra convolution (L-conv) as

$$Q[f](\boldsymbol{x}) = W^0 \left[ I + \bar{\epsilon}^i \hat{L}_i \right] f(\boldsymbol{x})$$
$$= W^0 \left[ I + \bar{\epsilon}^i [gL_i\boldsymbol{x}_0]^\alpha \partial_\alpha \right] f(\boldsymbol{x}) \tag{10}$$

Being an expansion of G-conv, L-conv inherits the equivariance of G-conv, as we show next.

**Proposition 1** (Equivariance of L-conv). *With assumptions above, L-conv is equivariant under $G$.*

*Proof:* First, note that the components of $\hat{L}_i$ transform as $[\hat{L}_i(v\boldsymbol{x})]^\alpha = [vgL_i\boldsymbol{x}_0]^\alpha = v_\beta^\alpha \hat{L}_i(\boldsymbol{x})^\beta$, while the partial transforms as $\partial/\partial[v\boldsymbol{x}]^\alpha = [v^{-1}]_\alpha^\gamma \partial_\gamma$. As a result in $\hat{L}_i = [gL_i\boldsymbol{x}_0]^\alpha \partial_\alpha$ all factors of $v$ cancel, meaning for $v \in G$, $\hat{L}_i(v\boldsymbol{x}) = \hat{L}_i(\boldsymbol{x})$. This is because of the fact that $\hat{L}_i \in T\mathcal{S}$ is a vector field (i.e. 1-tensor) and, thus, invariant under change of basis. Plugging into equation 10, for $w \in G$

$$w \cdot Q[f](\boldsymbol{x}) = Q[f](w^{-1}\boldsymbol{x}) = W^0 \left[ I + \bar{\epsilon}^i \hat{L}_i(w^{-1}\boldsymbol{x}) \right] f(w^{-1}\boldsymbol{x})$$
$$= W^0 \left[ I + \bar{\epsilon}^i \hat{L}_i(g) \right] f(w^{-1}\boldsymbol{x}) = W^0 \left[ I + \bar{\epsilon}^i \hat{L}_i(g) \right] w \cdot f(\boldsymbol{x}) = Q[w \cdot f](\boldsymbol{x}) \tag{11}$$

which proves L-conv is equivariant. $\square$

**Examples.** Using equation 9 we can calculate L-conv for specific groups (details in SI A.2). For translations $G = T_n = (\mathbb{R}^n, +)$, we find the generators become simple partial derivatives $\hat{L}_i = \partial_i$ (SI A.2.2), yielding $f(\boldsymbol{x}) + \epsilon^\alpha \partial_\alpha f(\boldsymbol{x})$. For 2D rotations (SI A.2.1) the generator $\hat{L} \equiv (x\partial_y - y\partial_x) = \partial_\theta$, which is the angular momentum operator about the z-axis in quantum mechanics and field theories. For rotations with scaling, $G = SO(2) \times \mathbb{R}^+$, we have two $L_i$, one $\hat{L}_\theta = \partial_\theta$ from $so(2)$ and a scaling with $L_r = I$, yielding $\hat{L}_r = x\partial_x + y\partial_y = r\partial_r$. Next, we discuss the form of L-conv on discrete data.

### 3.1 Approximating G-conv using L-conv

L-conv can be used as a basic building block to construct G-conv with more general kernels. Figure 2 sketches the argument described here (see also SI A.1).

**Theorem 1** (G-conv from L-convs). *G-conv equation 3 can be approximated using L-conv layers.*

*Proof:* The procedure involves two steps, as illustrated in Fig. 2: 1) approximate the kernel using localized kernels as the $\delta_\eta$ in L-conv; 2) move the kernels towards identity using multiple L-conv layers. The following lemma outline the details. $\square$

**Lemma 1** (Approximating the kernel). *Let the kernel $\kappa : G \to \mathcal{F}' \otimes \mathcal{F}$ with $\int_G \|\kappa(g)\|^2 dg < \infty$ be continuously differentiable with $\|d\kappa(g)/dg\|^2 < \xi^2$, and with compact support over $G_0 \subset G$. Let $\kappa_k(g) = c_k \delta_\eta(u_k^{-1}g)$ be a set of $N$ kernels with support on an $\eta$ neighborhood of $u_k \in G$. Then there exist $c_k \in \mathcal{F}' \otimes \mathcal{F}$ and $u_k \in G$ such that $\tilde{\kappa} = \sum_{k=1}^N \kappa_k$ approximates $\kappa$, meaning $\int_G \|\kappa(g) - \tilde{\kappa}(g)\|^2 dg < \zeta^2$ for arbitrary small $\zeta \in \mathbb{R}_+$.*

*Proof:* See SI A.1 for details. The intuition is similar to the universal approximation theorem for neural networks (Hornik et al., 1989; Cybenko, 1989), only generalized to a group manifold instead of $\mathbb{R}$. Let $B_0$ be the set of $v_\epsilon = I + \epsilon^i L_i \in \mathfrak{g}$, with $\|\epsilon\|^2 < \eta^2$. Choose a set of $u_k \in G$ such that the neighborhoods $B_k = u_k B_0 \subset G$ cover the support $G_0$ of $\kappa$. The bound $\|d\kappa(g)/dg\|^2 < \xi^2$ means that on small enough neighborhoods $B_k \subset G$, for any two $u, v \in B_k$ we have $\|\kappa(u) - \kappa(v)\|^2 \leq \eta^2 \xi^2$, where $|G_0|$ is the volume of the support of $\kappa$. Hence, for $g \in B_k$, $\kappa(g)$ can be approximated with $\kappa_k(g) = \kappa(u_k)\delta_\eta(u_k^{-1} g)$, with normalized localized kernels $\delta_\eta(g)$, and any element $u_k \in B_k$. We show that the approximation error of using $\tilde{\kappa} = \sum_k \kappa_k$ to approximate $\kappa$ is bounded by $\int_G dg\|\kappa(g) - \tilde{\kappa}(g)\|^2 < |G_0|\eta^2 \xi^2$. Any desired error bound $\zeta$ can then be attained by choosing small enough $\eta$ for neighborhood sizes. $\qquad\square$

Thus, we can approximate a large class of kernels as $\kappa(g) \approx \sum_k \kappa_k(g)$ where the local kernels $\kappa_k(g) = c_k \delta_\eta(u_k^{-1} g)$ have support only on an $\eta$ neighborhood of $u_k \in G$. Here $c_k \in \mathbb{R}^{m'} \otimes \mathbb{R}^m$ are constants and $\delta_\eta(u)$ is as in equation 8. Using this, G-conv equation 3 becomes

$$[\kappa \star f](g) = \sum_k c_k \int dv \delta_\eta(u_k^{-1} v) f(gv) = \sum_k c_k [\delta_\eta \star f](gu_k). \tag{12}$$

The kernels $\kappa_k$ are localized around $u_k$, whereas in L-conv the kernel is around identity. We can compose L-conv layers to move $\kappa_k$ from $u_k$ to identity.

**Lemma 2** (Moving kernels to identity). *$\kappa_k$ can be moved near identity using a multilayer L-conv.*

*Proof:* In equation 12, write $u_k = v_\epsilon u_k'$, with $v_\epsilon = I + \epsilon^i L_i \in \mathfrak{g}$. Using the definition equation 10 an L-conv layer $Q_\epsilon = I - \epsilon^i \hat{L}_i$ performs a first order Taylor expansion (SI A.1) and so $Q_\epsilon[\delta_\eta](u_k'^{-1} v) = \delta_\eta(u_k^{-1} v) + O(\epsilon^2)$. Thus, applying one L-conv layer moves the localized kernel along $v_\epsilon$ on $G$. Writing $u_k$ as the product of a set of small group elements $u_k = \prod_{a=1}^p v_a$, with $v_a = I + \epsilon_a^i L_i \in \mathfrak{g}$. Defining L-conv layers $Q_a = I - \epsilon_a^i \hat{L}_i$, we can write

$$\kappa_k(g) \approx c_k Q_p \circ \cdots \circ Q_1 \circ \delta_\eta(g) \tag{13}$$

meaning $\kappa_k$ localized around $u_k$ can be written as a $p$ layer L-conv acting on a kernel $\delta_\eta(g)$, localized around the identity of the group. With $\|\epsilon_a\| < \eta$, the error in $u_k$ is $O(\eta^{p+1})$. $\qquad\square$

Thus, we conclude that any G-conv equation 3 can be approximated by multilayer L-conv. Furthermore, for compact $G$, using the theorem in Kondor & Trivedi (2018), we can show that any equivariant feedforward neural network can be approximated using multilayer L-conv with nonlinearities.

**Equivariance of nonlinearity.** Pointwise nonlinearities give equivariant maps between scalar feature maps. To see this, let $\sigma : \mathbb{R} \to \mathbb{R}$. We extend $\sigma : \mathcal{F} \to \mathcal{F}$ by applying $\sigma$ component-wise. Let $f : \mathcal{S} \to \mathcal{F}$ be a scalar feature map (i.e., $g \cdot f(\boldsymbol{x}) = f(g^{-1}\boldsymbol{x})$). Then

$$g \cdot (\sigma \circ (f))(\boldsymbol{x}) = \sigma \circ (f)(g^{-1}\boldsymbol{x}) = \sigma \circ (g \cdot f)(\boldsymbol{x}).$$

Since the composition of equivariant maps is equivariant, given equivariant linear mapping $Q : \mathcal{F}^{\mathcal{S}} \to \mathcal{F}'^{\mathcal{S}}$ (i.e. $g \cdot Q[f] = Q[g \cdot f]$), the layer $f \mapsto \sigma \circ Q[f]$ is equivariant. Hence we have the corollary:

**Corollary 1.** *Assume $G$ is compact and acts on $\mathcal{S}$ transitively. Then any equivariant feedforward neural network (FNN) can be approximated using multilayer L-conv with point-wise nonlinearities.*

*Proof:* A FNN is defined as $\sigma_p \circ F_p[\cdots [\sigma_1 \circ F_1[f]](\boldsymbol{x})$ where $F_k$ are linear and $\sigma_k$ are point-wise nonlinearities. By Theorem 1 of Kondor & Trivedi (2018), any linear layer in the equivariant FNN is a G-conv, which by Theorem 1 can be approximated by multilayer L-conv. Therefore, multilayer L-conv with nonlinearity can approximate any equivariant FNN. $\qquad\square$

Finally, to our knowledge it is not known whether *every equivariant function* can be approximated by equivariant FNN for a Lie group $G$. Hence, the corollary above is *not* a universal approximation theorem for equivariant scalar functions in terms of L-conv. However, it does show that multilayer $L$-conv is equally expressive as other equivariant networks. Next, we discuss implementation details.

## 4    Discretized space and implementation: the tensor notation

In many datasets, such as images, $f(\boldsymbol{x})$ is not given as continuous function, but rather as a discrete array, with $\mathcal{S} = \{\boldsymbol{x}_0, \ldots \boldsymbol{x}_{d-1}\}$ containing $d$ points. Each $\boldsymbol{x}_\mu$ represents a coordinate in higher dimensional space, e.g. on a $10 \times 10$ image, $\boldsymbol{x}_0$ is $(x, y) = (0, 0)$ point and $\boldsymbol{x}_{99}$ is $(x, y) = (9, 9)$.

**Feature maps and group action** In the tensor notation, we encode $\boldsymbol{x}_\mu \in \mathcal{S}$ as the canonical basis (one-hot) vectors in $\boldsymbol{x}_\mu \in \mathbb{R}^d$ with $[\boldsymbol{x}_\mu]_\nu = \delta_{\mu\nu}$ (Kronecker delta), e.g. $\boldsymbol{x}_0 = (1, 0, \ldots, 0)$. The features become $\boldsymbol{f} \in \mathcal{F} = \mathbb{R}^d \otimes \mathbb{R}^m$, meaning $d \times m$ tensors, with $f(\boldsymbol{x}_\mu) = \boldsymbol{x}_\mu^T \boldsymbol{f} = \boldsymbol{f}_\mu$. Although $\mathcal{S}$ is discrete, the group acting on $\mathcal{F}$ can be continuous (e.g. image rotations). Any $G \subseteq \mathrm{GL}_d(\mathbb{R})$ of the general linear group (invertible $d \times d$ matrices) acts on $\boldsymbol{x}_\mu \in \mathbb{R}^d$ and $\boldsymbol{f} \in \mathcal{F}$. We define $f(g \cdot \boldsymbol{x}_\mu) = \boldsymbol{x}_\mu^T g^T \boldsymbol{f}, \forall g \in G$, so that for $w \in G$ we have

$$w \cdot f(\boldsymbol{x}_\mu) = f(w^{-1} \cdot \boldsymbol{x}_\mu) = \boldsymbol{x}_\mu^T w^{-1T} \boldsymbol{f} = [w^{-1} \boldsymbol{x}_\mu]^T \boldsymbol{f} \tag{14}$$

Dropping the position $\boldsymbol{x}_\mu$, the transformed features are matrix product $w \cdot \boldsymbol{f} = w^{-1T} \boldsymbol{f}$. We can write G-conv in this notation (SI B). Similarly, we can rewrite L-conv equation 8 in the tensor notation. Defining $v_\epsilon = I + \bar{\epsilon}^i L_i$

$$Q[\boldsymbol{f}](g) = W^0 f \left( g \left( I + \bar{\epsilon}^i L_i \right) \right) = \boldsymbol{x}_0^T \left( I + \bar{\epsilon}^i L_i \right)^T g^T \boldsymbol{f} W^{0T}$$
$$= \left( \boldsymbol{x} + \bar{\epsilon}^i [g L_i \boldsymbol{x}_0] \right)^T \boldsymbol{f} W^{0T}. \tag{15}$$

Here, $\hat{L}_i = g L_i \boldsymbol{x}_0$ is exactly the matrix analogue of pushforward vector field $\hat{L}_i$ in equation 9. The equivariance of L-conv in tensor notation is again evident from the $g^T \boldsymbol{f}$, resulting in

$$Q[w \cdot \boldsymbol{f}](g) = \boldsymbol{x}_0^T v_\epsilon^T g^T w^{-1T} \boldsymbol{f} W^{0T} = Q[\boldsymbol{f}](w^{-1}g) = w \cdot Q[\boldsymbol{f}](g) \tag{16}$$

**Tensor L-conv layer implementation** The discrete space L-conv equation 15 can be rewritten using the global Lie algebra basis $\hat{L}_i$

$$Q[\boldsymbol{f}] = \left( \boldsymbol{f} + \hat{L}_i \boldsymbol{f} \bar{\epsilon}^i \right) W^{0T}, \qquad Q[\boldsymbol{f}]_\mu^a = \boldsymbol{f}_\mu^b [W^{0T}]_b^a + [\hat{L}_i]_\mu^\nu \boldsymbol{f}_\nu^c \left[ W^i \right]_c^a \tag{17}$$

Where $W^i = W^0 \bar{\epsilon}^i$, $W^0 \in \mathbb{R}^{m_{in}} \otimes \mathbb{R}^{m_{out}}$ and $\bar{\epsilon}^i \in \mathbb{R}^{m_{in}} \otimes \mathbb{R}^{m_{in}}$ are trainable weights. The $\hat{L}_i$ can be either inserted as inductive bias or they can be learned to discover symmetries.

To implement L-conv, note that the formula of equation 17 is quite similar to a Graph Convolutional Network (GCN) (Kipf & Welling, 2016). For each $i$, the shared convolutional weights are $\bar{\epsilon}^i W^{0T}$ and the aggregation function of the GCN, a function of the graph adjacency matrix, is $\hat{L}_i$ in L-conv. Thus, L-conv can be implemented as GCN modules for each $\hat{L}_i$, plus a residual connection for the $\boldsymbol{f} W^{0T}$ term.

Figure 3 shows the schematic of the L-conv layer. In a naive implementation, $\hat{L}_i$ can be general $d \times d$ matrices. However, being vector fields generated by the Lie algebra, $\hat{L}_i$ has a more constrained structure which allows them to be encoded and learned using much fewer parameters than a $d \times d$ matrix. Specifically, encoding the topology of $\mathcal{S}$ as a graph (see SI B.1), the incidence matrix replaces partial derivatives (Schaub et al., 2020) in equation 9 and the $L_i$ become weighting of the edges. This weighting is similar to

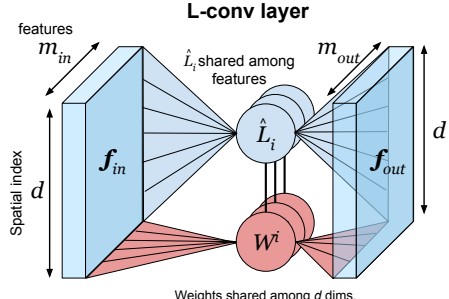

**L-conv layer**

Figure 3: L-conv layer architecture. $L_i$ only act on the $d$ flattened spatial dimensions, and $W^i$ only act on the $m_{in}$ input features and returns $m_{out}$ output features. For each $i$, L-conv is analogous to a GCN with $d$ nodes and $m_{in}$ features.

Gauge Equivariant Mesh (GEM) CNN (Cohen et al., 2019a). Indeed, in L-conv the lift $\boldsymbol{x}_\mu = g_\mu \boldsymbol{x}_0$ fixes the gauge by mapping neighbors of $\boldsymbol{x}_0$ to neighbors of $\boldsymbol{x}_\mu$. Changing how the discrete $\mathcal{S}$ samples an underlying continuous space will change $g_\mu$ and hence the gauge.

**Choosing the number of $L_i$.** Beside the width of $W^0$ and $\bar{\epsilon}^i$, the number $n_L$ of $L_i$ is a hyperparameter in L-conv. For instance, if $\mathcal{S}$ is a discretization of $n$ dimensional space the symmetry group is likely $G \subset \mathrm{GL}_n(\mathbb{R}) \ltimes T_n$, with $n_L \sim O(n^2)$. Note that $n_L$ is independent of the size $d$ of the discretized space (e.g. number of pixels) and generally $n^2 \ll d$. Choosing $n_L$ larger than the true number of $L_i$ only results in an over-complete basis and shouldn't be a problem. We conducted small controlled experiments to verify how multilayer L-conv approximates G-conv (SI C).

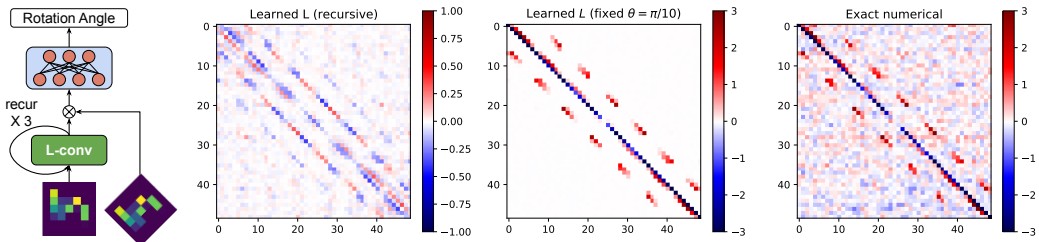

Figure 4: **Learning the infinitesimal generator of** $SO(2)$ Left shows the architecture for learning rotation angles between pairs of images (SI C.3). Next to it is the $L$ learned using recursive L-conv in this experiment. Middle $L$ is learned using a fixed small rotation angle $\theta = \pi/10$, and right shows $L$ found using the numeric solution from the data.

**Learning symmetries using L-conv.** Rao & Ruderman (1999) introduced a basic version of L-conv and showed that it can learn 1D translation and 2D rotation. We conducted experiments to learn large rotation angle between two images (SI C), shown in Fig. 4. Left shows the architecture for learning the rotation angles between a pair of $7 \times 7$ random images $\boldsymbol{f}$ and $R(\theta)\boldsymbol{f}$ with $\theta \in [0, \pi/3)$. Second left is the learned $L \in SO(2)$ using 3 recursive layer L-conv. Middle is the $L$ learned using L-conv with fixed small rotation angle $\theta = \pi/10$ (SI C.2) and right is the exact solution $R = (YX^T)(X^TX)^{-1}$. While the middle $L$ is less noisy, it does not capture weights beyond first neighbors of each pixel. (also see SI C for a discussion on symmetry discovery literature.)

L-conv can potentially replace other equivariant layers in a neural network. We conducted limited experiments for this on small image datasets (SI D). L-conv allows one to look for potential symmetries in data which may have been scrambled or harbors hidden symmetries.

## 5 Relation to other architectures

**CNN.** This is a special case of expressing G-conv as L-conv when the group is continuous 1D translations. The arguments here generalize trivially to higher dimensions. Rao & Ruderman (1999, sec. 4) used the Shannon-Whittaker Interpolation (Whitaker, 1915) to define continuous translation on periodic 1D arrays as $\boldsymbol{f}'_\rho = g(z)^\nu_\rho \boldsymbol{f}_\nu$. Here $g(z)^\nu_\rho = \frac{1}{d} \sum_{p=-d/2}^{d/2} \cos\left(\frac{2\pi p}{d}(z + \rho - \nu)\right)$ approximates the shift operator for continuous $z$. These $g(z)$ form a 1D translation group $G$ as $g(w)g(z) = g(w+z)$ with $g(0)^\nu_\rho = \delta^\nu_\rho$. For any $z = \mu \in \mathbb{Z}$, $g_\mu = g(z = \mu)$ are circulant matrices that shift by $\mu$ as $[g_\mu]^\rho_\nu = \delta^\rho_{\nu-\mu}$. Thus, a 1D CNN with kernel size $k$ can be written suing $g_\mu$ as

$$F(\boldsymbol{f})^a_\nu = \sigma\left(\sum_{\mu=0}^{k} \boldsymbol{f}^c_{\nu-\mu}[W^\mu]^a_c + b^a\right) = \sigma\left(\sum_{\mu=0}^{k}[g_\mu \boldsymbol{f}]^c_\nu [W^\mu]^a_c + b^a\right) \quad (18)$$

where $W, b$ are the filter weights and biases. $g_\mu$ can be approximated using the Lie algebra and written as multi-layer L-conv as in sec. 3.1. Using $g(0)^\nu_\rho \approx \delta(\rho - \nu)$, the single Lie algebra basis $[\hat{L}]_0 = \partial_z g(z)|_{z\to 0}$, acts as $\hat{L}f(z) \approx -\partial_z f(z)$ (because $\int \partial_z \delta(z - \nu)f(z) = -\partial_\nu f(\nu)$). Its components are $\hat{L}^\nu_\rho = L(\rho - \nu) = \sum_p \frac{2\pi p}{d^2} \sin\left(\frac{2\pi p}{d}(\rho - \nu)\right)$, which are also circulant due to the $(\rho - \nu)$ dependence. Hence, $[\hat{L}\boldsymbol{f}]_\rho = \sum_\nu L(\rho - \nu)\boldsymbol{f}_\nu = [L \star \boldsymbol{f}]_\nu$ is a convolution. Rao & Ruderman (1999) already showed that this $\hat{L}$ can reproduce finite discrete shifts $g_\mu$ used in CNN. They used a primitive version of L-conv with $g_\mu = (I + \epsilon\hat{L})^N$. Thus, L-conv can approximate 1D CNN. This result generalizes easily to higher dimensions.

**Graph Convolutional Network (GCN).** Let $\boldsymbol{A}$ be the adjacency matrix of a graph. In equation 17 if $\hat{L}_i = h(\boldsymbol{A})$, such as $\hat{L}_i = \boldsymbol{D}^{-1/2}\boldsymbol{A}\boldsymbol{D}^{-1/2}$, we obtain a GCN (Kipf & Welling, 2016) ($\boldsymbol{D}_{\mu\nu} = \delta_{\mu\nu} \sum_\rho \boldsymbol{A}_{\mu\rho}$ being the degree matrix). So in the special case where all neighbors of each node $<\mu>$ have the same edge weight, meaning $[\hat{L}_i]^\nu_\mu = [\hat{L}_i]^\rho_\mu, \forall \nu, \rho \in <\mu>$, equation 8 is uniformly aggregating over neighbors and L-conv reduces to a GCN. Note that this similarity is not just superficial. In GCN $h(\boldsymbol{A}) = \hat{L}$ is in fact a Lie algebra basis. When $\hat{L} = h(\boldsymbol{A})$, the vector field is the

flow of isotropic diffusion $d\boldsymbol{f}/dt = h(\boldsymbol{A})\boldsymbol{f}$ from each node to its neighbors. This vector field defines one parameter Lie group with elements $g(t) = \exp[h(\boldsymbol{A})t]$. Hence, L-conv for flow groups with a single generator are GCN. These flow groups include Hamiltonian flows and other linear dynamical systems. The main difference between L-conv and GCN is that L-conv can assign a different weight to each neighbor of the same node, similar to GEM-CNN (Cohen et al., 2019a) with a fixed gauge set by $g_\mu$. Next, we discuss the mathematical properties of the loss functions for L-conv.

## 6 Group invariant loss

Loss functions of equivariant networks are rarely discussed. Yet, recent work by Kunin et al. (2020) showed the existence of symmetry directions in the loss landscape. To understand how the symmetry generators in L-conv manifest themselves in the loss landscape, we work out the explicit example of a mean square error (MSE) loss. Because $G$ is the symmetry group, $f$ and $g \cdot f$ should result in the same optimal parameters. Hence, the minima of the loss function need to be *group invariant*. One way to satisfy this is for the loss itself to be group invariant, which can be constructed by integrating over $G$ (global pooling (Bronstein et al., 2021)). A function $I = \int_G dg F(g)$ is $G$-invariant (SI A.3). We can also change the integration to $\int_{\mathcal{S}} d^n x$ by change of variable $dg/d\boldsymbol{x}$ (see SI A.3 for discussion on stabilizers).

**MSE loss and Field Theory.** The MSE is given by $I = \sum_n \int_G dg \|Q[f_n](g)\|^2$, where $f_n$ are data samples and $Q[f]$ is L-conv or another $G$-equivariant function. In supervised learning the input is a pair $f_n, y_n$. $G$ can also act on the labels $y_n$. We assme that $y_n$ are either also scalar features $y_n : \mathcal{S} \to \mathbb{R}^{m_y}$ with a group action $g \cdot y_n(\boldsymbol{x}) = y_n(g^{-1}\boldsymbol{x})$ (e.g. $f_n$ and $y_n$ are both images), or that $y_n$ are categorical. In the latter case $g \cdot y_n = y_n$ because the only representations of a continuous $G$ on a discrete set are constant. We can concatenate the inputs to $\phi_n \equiv [f_n|y_n]$ with a well-defined $G$ action $g \cdot \phi_n = [g \cdot f_n|g \cdot y_n]$. The collection of combined inputs $\Phi = (\phi_1, \ldots, \phi_N)^T$ is an $(m + m_y) \times N$ matrix. Using equations 8 and 9, the MSE loss with parameters $W = \{W^0, \bar{\epsilon}\}$ becomes (SI A.3.1)

$$I[\Phi; W] = \int_G dg \mathcal{L}[\Phi; W] = \int_G dg \left\| W^0 \left[ I + \bar{\epsilon}^i [\hat{L}_i]^\alpha \partial_\alpha \right] \Phi(g) \right\|^2$$

$$= \int_{\mathcal{S}} \frac{d^n x}{\left| \frac{\partial x}{\partial g} \right|} \left[ \Phi^T \mathbf{m}_2 \Phi + \partial_\alpha \Phi^T \mathbf{h}^{\alpha\beta} \partial_\beta \Phi + [\hat{L}_i]^\alpha \partial_\alpha \left( \Phi^T \mathbf{v}^i \Phi \right) \right] \tag{19}$$

Equation 19 generalizes the free field theories in physics (Polyakov, 2018). Here $\left| \frac{\partial x}{\partial g} \right|$ is the determinant of the Jacobian, $W^i = W^0 \bar{\epsilon}^i$ and

$$\mathbf{m}_2 = W^{0T} W^0, \qquad \mathbf{h}^{\alpha\beta}(\boldsymbol{x}) = \bar{\epsilon}^{iT} \mathbf{m}_2 \bar{\epsilon}^j [\hat{L}_i]^\alpha [\hat{L}_j]^\beta, \qquad \mathbf{v}^i = \mathbf{m}_2 \bar{\epsilon}^i. \tag{20}$$

Note that $\mathbf{h}$ has feature space indices via $[\bar{\epsilon}^{iT} \mathbf{m}_2 \bar{\epsilon}^j]_{ab}$, with index symmetry $\mathbf{h}^{\alpha\beta}_{ab} = \mathbf{h}^{\beta\alpha}_{ba}$. When $\mathcal{F} = \mathbb{R}$ (i.e. $f$ is a 1D scalar), $\mathbf{h}^{\alpha\beta}$ becomes a a Riemannian metric for $\mathcal{S}$. In general $\mathbf{h}$ combines a 2-tensor $\mathbf{h}_{ab} = \mathbf{h}^{\alpha\beta}_{ab} \partial_\alpha \partial_\beta \in T\mathcal{S} \otimes T\mathcal{S}$ with an inner product $h^T \mathbf{h}^{\alpha\beta} f$ on the feature space $\mathcal{F}$.

In field theory, the motivation is to preserve spatial symmetries for the metric $\mathbf{h}$. In equation 19, $\mathbf{h}$ transforms equivariantly as a 2-tensor $v \cdot \mathbf{h}^{\alpha\beta} = [v^{-1}]^\alpha_\rho [v^{-1}]^\beta_\gamma \mathbf{h}^{\rho\gamma}(\boldsymbol{x})$ for $v \in G$ (SI A.3). The last term in equation 19 vanishes for many groups (SI A.3) and it is also absent in physics.

**Robustness and Euler-Lagrange Equation.** Equivariant neural networks are more robust. To check this, we can quantify how the network would perform for an input $\phi' = \phi + \delta\phi$ which adds a small random perturbation $\delta\phi$ to a data point $\phi$. Robustness to such perturbation would mean that, for optimal parameters $W^*$, the loss function would not change, i.e. $I[\phi'; W^*] = I[\phi; W^*]$, requiring $I$ to be minimized around real data points $\phi$.

This can be cast as a variational equation $\delta I[\phi; W^*] = 0$, which yield the familiar Euler-Lagrange (EL) equation (SI A.4). Therefore, for an equivariant network to be robust, i.e. $\delta I[\phi; W^*]/\delta\phi = 0$, we would require the data points $\phi$ to satisfy the EL equations for optimal parameters $W^*$:

$$\text{Robustness to random noise} \iff \text{EL:} \quad \frac{\partial \mathcal{L}}{\partial \phi^b} - \partial_\alpha \frac{\partial \mathcal{L}}{\partial (\partial_\alpha \phi^b)} = 0 \tag{21}$$

where the partial derivative terms appear because of the L-conv layer.

**Equivariance and Conservation laws.** Conserved currents, via Noether's theorem provide a way to find hidden symmetries (see also Kunin et al. (2020)). The idea is that the equivariance condition equation 2 can be written for the integrand of the loss, $\mathcal{L}[\phi, W]$. If we write the equivariance equation for infinitesimal $v_\epsilon$, we obtain a vector field which is divergence free. Since $G$ is the symmetry of the system, transforming an input $\phi \to w \cdot \phi$ by $w \in G$ the integrand should change equivariantly, meaning $\mathcal{L}[w \cdot \phi] = w \cdot \mathcal{L}[\phi]$. When robustness error is minimized as in equation 21, an infinitesimal $w \approx I + \eta^i L_i$, with $\delta\phi = \epsilon^i \hat{L}_i \phi$, results in a conserved current (SI A.4)

$$\text{Noether current: } J^\alpha = \frac{\partial \mathcal{L}}{\partial(\partial_\alpha \phi^b)}\delta\phi^b - \frac{\partial \mathcal{L}}{\partial \boldsymbol{x}^\alpha}\delta\boldsymbol{x}^\alpha, \qquad \delta I[\phi; W^*] = 0 \quad \Rightarrow \partial_\alpha J^\alpha = 0 \qquad (22)$$

The above equation shows that for equivariant networks with a given symmetry, the deviation in data along the symmetry direction ($\hat{L}_i$) yields a divergence free current $J^\alpha$, known as Noether current. It also provides an alternative means to discover symmetry generators $L_i$ by minimizing $\|\partial_\alpha J^\alpha\|$. Note that this Noether current is the "stress-energy" tensor, associated with space (or space-time) variations $\delta\boldsymbol{x}$ (Landau, 2013) (SI A.5). We can potentially design more general equivariant networks leading to other Noether currents.

# 7    Conclusion and Discussions

We propose the Lie algebra convolutional neural network (L-conv), an infinitesimal version of G-conv. L-conv layers do not require encoding irreps or discretizing the group, and can be combined to approximate *any* feedforward equivariant networks on compact groups. Additionally, L-conv's universal and simple structure allows us to discover symmetries from data. It is easy to implement, with a formula similar to GCN. We validated that L-conv can learn the correct Lie algebra basis in a synthetic experiment.

We discover several intriguing connections between L-conv and physics. Our derivation shows that equivariant neural networks based on L-conv lead to Noether's theorem and conservation laws. Conversely, we can also optimize Noether current to discover symmetries. Furthermore, the current equivariance formulation only pertains to "spatial symmetries" (i.e. $G$ acts on $\mathcal{S}$). In physics, more general "internal symmetries" are quite common (e.g. particle physics). We can potentially design more general equivariant networks with L-conv encoding such symmetries.

Our method also shed lights on scientific machine learning, especially for physical sciences. Physicists generally use simple polynomial forms for the Lagrangian, or the loss function. These "perturbative" Lagrangian lead to divergences in quantum field theory. However, it is believed the true Lagrangian is more complicated. Hence, more expressive L-conv based models can potentially provide more advanced ansatze for solving scientific problems.

## Acknowledgments and Disclosure of Funding

R. Walters is supported by a Postdoctoral Fellowship from the Roux Institute and NSF grants #2107256 and #2134178. This work was supported in part by the U. S. Army Research Office under Grant W911NF-20-1-0334, DOE ASCR 2493 and NSF Grant #2134274. N. Dehmamy and D. Wang were supported by the Air Force Office of Scientific Research under award number FA9550-19-1-0354.

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
