# OpenReview forum: "Automatic Symmetry Discovery with Lie Algebra Convolutional Network"
_NeurIPS.cc/2021/Conference — NeurIPS 2021 Poster_

### Official Review · Reviewer_rszC · 2021-07-04

**Rating:** 5
**Confidence:** 3

**Summary:**

The paper introduces an equivariant L-Conv layer, a building block that can be used to construct group equivariant layers. These L-conv layers are used to discover symmetries in data, and various connections between L-Conv and concepts in physics are made.


**Limitations And Societal Impact:**

The authors do not mention any limitations of their work. There are clear limitations are in the lack of clarity and organisation in the presentation of the paper, which is making it difficult to assess the limitations of the proposed methods and ideas in the paper.

**Main Review:**

The paper lacks focus and it is difficult to see what the main contribution of the paper is. Each contribution in the list of contributions in the Intro is presented poorly, which makes it difficult to assess the strenghts and weaknesses of L-Conv and its use for discovering symmetries from data.

- Majority of paper is on L-Conv, a new group equivariant layer, suggesting that this new methodology is the main focus and this is a methodology paper. If this were the case, there should be more details in the main paper about the practical benefits of L-Conv over baseline group equivariant layers such as G-conv, shown by experimental results. However all experimental results (except from a brief, insufficiently described result in Figure 3) are placed in the appendix, making the practicality of L-Conv difficult to assess. Furthermore, the results on images in Appendix D comparing to group equivariant baselines such as LieConv are disappointing - all models show test accuracy < 0.5, and LieConv seems to outperform L-Conv despite having much fewer parameters, except the rotated&scrambled image case which is a rather contrived scenario.

- The title of the paper, on the other hand, suggests that symmetry discovery is a main component of the paper. However there is very little description of this in the main paper (a single paragraph in Section 3.1), and due to the brevity it is difficult to understand how symmetry discovery is happening and how the reader is supposed to interpret the learned L matrix in Figure 3. The task in Figure 3 seems to be a task trying to predict rotation angles between two rotated versions of the same input, which I'd argue is not really an instance of 'symmetry discovery'. Symmetry discovery usually refers to the case when given a G-equivariant/invariant task with unknown G, one has to narrow down the set of possible groups G. However for the task in Figure 3, any model that solves the task has to be rotation equivariant, so there is no discovery of symmetry involved here. It would be more relevant/interesting to see what L_i is learned by L-Conv when used on a more standard invariant/equivariant task such as image classification or object localisation, to see if L-Conv can learn translational/rotational symmetry. Furthermore when L_i is learned, the number of lie algebra generators (the number of L_i learned) determines what groups can/cannot be learned, which seems to be a limitation of the method for symmetry discovery.

- Re connections between MSE loss function/generalisation error for L-Conv and concepts in Physics, it's not clear what are the insights gained from these connections. The paper merely points out these connections, but does not expand on what can be learned from such connections.

- The main paper reads to be a summary of the appendix, with considerable overlap between the main paper and the appendix (e.g. Sections 3&6 <-> Appendix A, Section 4 <-> Appendix B.1). I find myself having to read the appendix to make sense of the main paper - this is another example showing that the paper is disorganised, since the roles of the main paper and the supplementary material are reversed.

Introduction of L-Conv layer in Section 3 is confusing.

- Equation (6) is hard to follow. The equality in the first line seems to be from Taylor expansion (it'd be good to mention this explicitly for clarity), but the jump from line 1 to 2 in Equation (6) seems erroneous. Shouldn't the second line be W^0 f(g) + c \bar{epsilon}^i g L_i df/dg instead? Hence the last line of equation (6), where you seem to have gone from the Taylor expansion back to the analytic expression, seems unjustified.

- In Section 3, it's not clear at all how L-conv is implemented in practice (e.g. what delta_eta is and how W^0, \bar{epsilon}_i are computed), and information on this only appears at the end of page6, towards the end of the paper, at which point I found myself having to go back to Section 3 to make sense of it. This is an instance showing that the paper is poorly organised and presented.

- Section 3.1 on approximating G-Conv using L-conv doesn't seem to tell us anything about how L-Conv can be made to approximate G-Conv in practice. There is no information about how the local kernels kappa_k(g) should be chosen given a G-Conv kernel \kappa(g), and there is no detail about the error introduced by taking a finite number of L-Convs to approximate a G-Conv. The statement "L-Convs can approximate G-Conv" doesn't mean much if there is no analysis on the approximation error.

- tensor notation further makes things all the more confusing, and seems unnecessary since all "tensors" are just matrices or vectors.


**Time Spent Reviewing:**

6 hours

---

> ### Author Response · Authors · 2021-08-06
> **Clarifications and some reorganization**
>
> Thank you for your comments.
>
> > Majority of paper is on L-Conv ... there should be more details in the main paper about the practical benefits of L-Conv over baseline ... G-conv, shown by experimental results.
>
> Thank you, this is a methods and theory paper, attempting to simplify the core of group equivariant architectures.
> Our main contribution is to develop a rigorous theoretical framework for L-Conv. While empirical verification of practical benefits are also important, we believe that the precise theoretical connection between group convolution (G-conv) and multiple layers of Lie algebra convolution  (L-conv) is sufficiently meaningful to warrant its own paper.
>
> > However all experimental results (except from a brief, insufficiently described result in Figure 3) are placed in the appendix, making the practicality of L-Conv difficult to assess.
>
>
> We also would like to note that L-conv is not the best solution for all equivariant tasks, just as MLP is not the best solution depsite being a universal approximator. As Fig.12 shows, on the "Default" image datasets CNN outperforms L-conv while using fewer parameters. The message is that, if we have good inductive bias (e.g. an efficient implementation of a group equivariant model), that model can outperform L-conv, which is trying to learn the symmetry and solve the task at the same time. Where L-conv is beneficial is tasks where we have no knowledge of the possible symmetries.
>
> > Furthermore, the results on images in Appendix D ... rather contrived scenario.
>
> Our method cannot outperform baselines with known symmetry inductive biases.  Rotated & scrambled MNIST is, respectfully, a perfect benchmark for methods which can exploit discovered unknown symmetry.  This presents a data distribution which definitively has symmetry but which does not match the translational or rotational inductive biases of CNN or E(2)-CNN and cannot be easily programmed as an inductive bias into LieConv.  Real-world data which might match this could include radar scattering data or x-ray diffraction data which has rotational and translational symmetries within the underlying physics but are difficult to appreciate in their usually presented form
>
> > The title of the paper, on the other hand, suggests that symmetry discovery is a main component of the paper. However there is very little description of this in the main paper (a single paragraph in Section 3.1),
>
> L-conv is motivated by symmetry discovery without imposing assumptions about the group. When the $L_i$ are learnable, there is always implicit symmetry discovery when using L-conv. However, as our model is learning the Lie algebra, it does not necessarily output an easy interpretation of the corresponding Lie group G.
>
> > and due to the brevity it is difficult to understand how symmetry discovery is happening and how the reader is supposed to interpret the learned L matrix in Figure 3.
>
> Please refer to Appendix C.1 for the ground truth (Fig.7.a) and how to interpret $L$. Our learned $L$ has a cosine correlation of 0.71 with the ground truth $L$. Following your comment, we could move Fig.7 to the main text.
>
> > The task in Figure 3 seems to be a task trying to predict rotation angles between two rotated versions of the same input, which I'd argue is not really an instance of 'symmetry discovery'. Symmetry discovery usually refers to ... G-equivariant/invariant task with unknown G, one has to narrow down the set of possible groups G.  However... in Figure 3, any model that solves the task has to be rotation equivariant, so there is no discovery of symmetry involved here.
>
> We, respectfully, disagree. In Figure 3, we are learning both the $L_i$ (group generator) and the rotation angle simultaneously. In L-conv, symmetry discovery only requires learning the right $L_i$ without needing to narrow down the set of possible groups. Also, even though the initial, untrained L-conv is not rotation equivariant, it becomes equivariant once it learns the correct $L$ for rotations.
>
> > It would be more relevant/interesting to see what L_i is learned by L-Conv ... symmetry.
>
> We already experimented with the standard tasks in Appendix D and provided two very narrow experiments (Fig 3, and Appendix C), necessitated learning the correct $L$. with further experiments (Supp, Figs. 11 and 13) we proved that even with less parameters L-conv was able to beat CNN in rotated and scrambled images.
> Unlike the (concurrently developed) Augerino method (https://arxiv.org/pdf/2010.11882.pdf) which considers a very small space of potential symmetries (translations, rotations, reflections, etc.), there are some subtleties to validating our method on domains with previously known symmetries.  The dimension of our potential symmetry space is very large (>10K dimensional for MNIST) and thus the L_i we discover will not necessarily resemble the typical Lie algebra generators for, e.g., Lie(SO(2)).  Instead we usually discover a different basis encoding a larger group of symmetries which includes those generators in its span.
>
> > Furthermore when L_i is learned, the number of lie algebra generators (the number of L_i learned) [limitation]... symmetry discovery.
>
> The number of generator is a hyperparameter, which is hardly a limitation.
>
> > Re connections between MSE loss function/generalisation error for L-Conv and concepts in Physics, ... what can be learned from such connections.
>
> The key insight is that the physics machinery for symmetries can now be used in ML and ML can help advance the knowledge of physics. More concretely:
> 1) The simplest loss function for a single layer L-conv yields free field theory. So, more sophisticated ML models using L-conv may allow us to learn physical phenomena for which simple field theory Lagrangians are not a good fit (I can elaborate on this if requested).
> 2) The variational (Euler-Lagrange) equations provide a way to measure the robustness L-conv and other G-conv are to random noise. Violations of these equations may reveal directed, adversarial attacks.
> 3) The Noether current derived from the Lagrangian (loss function) can be used as check for equivariance of a model under a given symmetry.
>
> Such connections suggest exciting interplay between physics and ML, but given the page limit, we did not get to elaborate on it adequately (more discussion in Supp).
>
> > The main paper reads to be a summary of the appendix ... Section 3 is confusing.
>
> We will reorganize section 3  with more details on Eqn 6 and move up the implementation of L-conv in the main paper.
>
> > Equation (6) is hard to follow.... back to the analytic expression, seems unjustified.
>
> The jump in eq. 6 is correct (see eq. 25, Appendix A) as up to first order the analytic expression and the Taylor expansion match. Nevertheless, we can write everything in terms of the linear Taylor expansion. The proof of equivariance in the linear expansion requires one extra step showing that the derivative term is invariant under the group action. Concretely, for $v\in G$
> $$ \hat{L}(vg)= vgL_i \cdot \frac{d}{d(vg)}=\mathrm{Tr}\left[(vgL_i)^T v^{-1}\frac{d}{dg}\right] = gL_i \cdot \frac{d}{dg}= \hat{L}(g)$$
> (this is because $\hat{L}$ is a vector field and invariant under change of basis).
> We will  include some of Appendix A, eq. 25 into the main text. Also, your expression is the same as line 2 ($W^0=c$, when $\int \delta_\eta(g)dg=1$).
>
> > In Section 3... [L-conv implemented in practice]...poorly organised and presented.
>
> Thank you for the point. We will move implementaion earlier. We will move eq. 10 and a version of eq. 16 right after eq. 6, before sec. 3.1.
> The implementation of L-conv is in eq. 16. Specifically, $W^0$ and $\epsilon$ are learnable weights. $\delta_\eta$ is irrelevant for learning L-conv. As in the proof of universal approximation theorem for perceptrons, $\delta_eta$ is a function like sigmoid which shows any G-conv can be approximated by many L-convs. The point of L-conv is that we only need to learn $W^0$ and $\epsilon$ instead of integrating $\delta_\eta$ over $G$.
>
> > Section 3.1 on approximating G-Conv using L-conv ... L-Convs to approximate a G-Conv.
>
> Similar to universal approximation (UA) proofs, we prove that there exist weights for L-conv which can approximate any G-conv.  In practice, however, we expect that these weights will be learned from data and not analytically derived from a G-conv. We demonstrate this works empirically (See Fig. 3) The experiment of Fig. 3 is one example of an L-conv approximating G-conv on $SO(2)$. Like UA, the width and number of $\kappa_k(g)$ determines the approximation error. In practice, $\kappa_k(g)$ are not chosen. Only their integrals $W^0=\int_G dg \kappa(g)$ and $\bar{\epsilon}^i=\int_G dg \epsilon^i \kappa(g)$ are learned. Since we only learn the integrated variables $W^0$ and $\bar{\epsilon}^i$, we don't need to know $\kappa(g)$ because we don't need to integrate over $G$.
>
>
> > The statement "L-Convs can approximate G-Conv" doesn't mean much if there is no analysis on the approximation error.
>
> Our results show the approximating error can be made arbitrarily small given large enough hyperparameters. See Appendix C, Fig 6 for a simple experimental results regarding the 1D translation group. The approximation error estimation is identical to the universal approximation theorem. It's the error of approximating a given function $\kappa(g)$ with  a set of localized functions $\delta_\eta(g)$. As stated in line 128,   $\delta_\eta$ can be written as the sum of two sigmoid functions and the UA estimation errors translate identically to the G-conv with L-conv approximation.
>
> > tensor notation... confusing ... unnecessary since all "tensors" are just matrices or vectors.
>
> The tensor notation shows rigorously how L-conv can be implemented on discretized spaces. It is necessary for applying L-conv in many practical setting, from images to point-clouds to spin systems.

---

> > ### Comment · Reviewer_rszC · 2021-08-18
> > **Response to rebuttal**
> >
> > Thank you for the clarifications. Some of my concerns have been addressed, so I will raise my score accordingly, but other concerns still remain.
> >
> > > Where L-conv is beneficial is tasks where we have no knowledge of the possible symmetries.
> >
> > I appreciate that LieConv is intended for use in cases where the symmetry for the problem at hand is unknown, hence the use of Rotated & scrambled MNIST as a benchmark. But the scrambling would destroy local structure, hence the CNN is bound to give poor results. A more suitable baseline would be MLPs here.
> >
> > > We, respectfully, disagree. In Figure 3, we are learning both the Li (group generator) and the rotation angle simultaneously. In L-conv, symmetry discovery only requires learning the right Li without needing to narrow down the set of possible groups. Also, even though the initial, untrained L-conv is not rotation equivariant, it becomes equivariant once it learns the correct L for rotations.
> >
> > I'm still not convinced that this task can be attributed to symmetry discovery. Wouldn't traditional Conv instead of L-Conv also be able to solve the task given enough data? The Conv at initialisation won't be rotation equivariant, but I would expect it to eventually learn to be equivariant (at least for the range of angles seen during training) via supervision as it learns to predict the correct rotation angle. But we wouldn't say that Convs are suitable for symmetry discovery. To show evidence for symmetry discovery, it might make sense to show errors for out-of-training-distribution angles, and see if L-conv can generalise (while showing that Convs would fail).
> >
> > >The number of generator is a hyperparameter, which is hardly a limitation.
> >
> > I think you misunderstood my point. My point was that the group of symmetries that is learned by L-Conv could be sensitive to this choice of hyperparameter. Can you provide guidance on how to choose this hyperparameter, or evidence that the results are insensitive to choice of this hyperparameter?
> >
> > >The jump in eq. 6 is correct (see eq. 25, Appendix A) as up to first order the analytic expression and the Taylor expansion match.
> >
> > If I understood correctly, it seems like you're using another Taylor expansion on $W^0$ to include a factor of $W^0$ for the second term, and then ignoring the $O(\epsilon^2)$ terms. However this step is quite unclear and doesn't seem rigorous because you seem to be multiplying first-order epsilon terms that are inside separate integrals, and treating this product as a second order term i.e. $\int O(\epsilon) d\epsilon \int O(\epsilon) d\epsilon \neq O(\epsilon^2)$. I think the paper needs a clearer derivation for the last equality of Equation (25) in Appendix A.

---

> > > ### Author Response · Authors · 2021-08-20
> > > **MLP baselines, symmetry discovery, number of $L_i$, Eq. 6**
> > >
> > > >Thank you for the clarifications. Some of my concerns have been addressed, so I will raise my score accordingly, but other concerns still remain.
> > >
> > > Thank you. We will do our best to address your concerns.
> > >
> > > >> Where L-conv is beneficial is tasks where we have no knowledge of the possible symmetries.
> > >
> > > >I appreciate that LieConv is intended for use in cases where the symmetry for the problem at hand is unknown, hence the use of Rotated & scrambled MNIST as a benchmark. But the scrambling would destroy local structure, hence the CNN is bound to give poor results. A more suitable baseline would be MLPs here.
> > >
> > > Yes, based on the same intuition, we included three versions of MLP in Figure 10 as baselines. "FC(~Lconv)" is an MLP which has the same number of layers and width as the Lconvs in the experiments (one layer for $L_i$, one for $W^0$ and $W^i=W^0\bar{\epsilon}^i$ for each Lconv layer), but due to lack of weight sharing it has more parameters  (Fig. 10, right, red bars). "FC(shallow)" is a single hidden layer MLP with parameter count matching the Lconv experiments. In some cases, such as MNIST, MLP indeed outperforms  CNN on scrambled-rotated data. But on CIFAR, CNN is better. Thus, despite the fact that scrambling destroys the local structure, CNN is not bound to give poor results. We want to emphasize that the scrambling does not affect L-conv, as its performance on rotated data is similar to scrambled-rotated.
> > > In fact, the data might be in a format that cannot be easily cast into a 2D image and the performance of L-conv would remain the same, similar to MLP.
> > >
> > >
> > >
> > > >> We, respectfully, disagree. In Figure 3, we are learning both the $L_i$ (group generator) and the rotation angle simultaneously. ...
> > >
> > > > I'm still not convinced that this task can be attributed to symmetry discovery. Wouldn't traditional Conv instead of L-Conv also be able to solve the task given enough data? ... To show evidence for symmetry discovery, it might make sense to show errors for out-of-training-distribution angles, and see if L-conv can generalise (while showing that Convs would fail).
> > >
> > >
> > > While it is true that a traditional Conv can solve the task given enough data, it is only learning the symmetry implicitly. In contrast, L-Conv learns to discover the symmetry explicitly by design. We can validate the learned symmetry group (Lie algebra basis) and compare it with the ground truth (Fig. 7). We can further look into out of sample predictions, or fix the learned $L$ and only train the $W$ and MLP to predict out of sample angles, to see if the learned $L$ works better than random $L$.
> > >
> > > >> The number of generator is a hyperparameter, which is hardly a limitation.
> > >
> > > > I think you misunderstood my point. My point was that the group of symmetries that is learned by L-Conv could be sensitive to this choice of hyperparameter. Can you provide guidance on how to choose this hyperparameter, or evidence that the results are insensitive to choice of this hyperparameter?
> > >
> > > Thank you for the clarification and the good suggestion. This hyperparameter is determined by the spatial symmetry group. Some intuition for choosing the number of $L_i$, $n_L$, can be as follows. If the base space $\mathcal{S}$ (before any discretization) is an $n$ dimensional vector space (and assuming the origin $\vec{x}=(0,...0)$ is known) the symmetry group is expected to be a subgroup of $\mathrm{GL}_n(\mathbb{R})\ltimes T_n$, with $T_n$ being the translation group. For example, images are features over a 2D space, meaning $n=2$.
> > > Since $T_n$ has $n$ Lie algebra generators and $\mathrm{gl}_n(\mathbb{R})$ has $n^2$, we expect $n_L\leq n+n^2$. For instance, the symmetry generators used by Augerino and LieConv are a subset of these $n_L$ generators.
> > > But these $n_L$ generators also include Lorentzian symmetries of Special Relativity, among others.
> > >
> > > If the space is more general than the vector space described above (e.g. a curved manifold), potentially any subgroup of its diffeomorphisms can be $G$, a possibly infinite dimensional group. In the tensor notation where the space is discretized to $d$ nodes (e.g. $d=\mathrm{height}\times\mathrm{width}$ pixels in images), the largest group that can appear in the data is $\mathrm{GL}_d(\mathbb{R})$, hence $n_L\leq d^2$. But generally we expect $n_L$ to be much smaller than $d^2$ because the resolution of the discretization usually does not reflect a true property of the system and the data may live on a manifold of much lower dimensionality.  For example, as mentioned for images, although the image may have $d=H*W$ pixels, the underlying space before discretization is 2D and so $GL_2(\mathbb{R})\ltimes T_2$ is likely enough to accommodate all spatial symmetries.
> > >
> > > Finally, having redundant $L_i$, which are not linearly independent, does not impact the performance of L-conv. For two bases $L_i$ and $L'_j$, even if their numbers differ, as long as $\bar{\epsilon}^iL_i= {\bar{\epsilon}'}^j L'_j$ they'll have the same performance. We can reduce such redundancy using regularizers enforcing linear independence.
> > >
> > > >> The jump in eq. 6 is correct (see eq. 25, Appendix A) as up to first order the analytic expression and the Taylor expansion match.
> > >
> > > > If I understood correctly, it seems like you're using another Taylor expansion on $W^0$ to include a factor of $W^0$ for the second term, and then ignoring the $O(\epsilon^2)$ terms.
> > > However this step is quite unclear and doesn't seem rigorous because you seem to be multiplying first-order epsilon terms that are inside separate integrals, and treating this product as a second order term i.e.
> > > $\int O(\epsilon) d\epsilon \int O(\epsilon) d\epsilon \ne O(\epsilon^2)$.
> > > I think the paper needs a clearer derivation for the last equality of Equation (25) in Appendix A.
> > >
> > > We will work to clarify Eq. 6. Note that each L-conv layer has only one integral, so there is no product of two integrals.
> > > We are only expanding $f(gv_\epsilon)$, and not $\kappa_I(v_\epsilon)=c\delta_\eta(v_\epsilon)$ which are distributions over $G$, with $\int d\epsilon \delta_\eta = 1$.
> > > The point is, the integration variable $\epsilon$ itself is not small, but the support of $\kappa_I(I+\epsilon^i L_i)$ is bounded by a ball of radius  $\eta$, which is small, and therefore $\bar{\epsilon}^k=\int d\epsilon \delta_\eta(I+\epsilon^iL_i) \epsilon^k$ is small.
> > > Consider an order $p$ function $\phi(\epsilon)$. From $|\epsilon| < \eta$, we have $|\phi(\epsilon)| < \eta^p C$ (for some const $C$).
> > > Substituting this bound into the integral we have
> > > $$ \left|\int d\epsilon \delta_\eta (I+\epsilon^i L_i) \phi(\epsilon)\right| \leq \int \left| d\epsilon \delta_\eta (I+\epsilon^i L_i) \phi(\epsilon)\right| < \int \left| d\epsilon \delta_\eta (I+\epsilon^i L_i)\eta^p C\right| \leq \eta^p C \int \left| d\epsilon \delta_\eta (I+\epsilon^i L_i)\right| \leq \eta^p C$$
> > >
> > > When $f(gv_\epsilon)=f(g+\epsilon^i gL_i) $ is expanded to first order in $\epsilon$, it yields
> > > $$ f(g+\epsilon^i gL_i)= f(g)+\epsilon^i gL_i\cdot {df(g)\over dg} +O(\epsilon^2) $$
> > > Note the $f(g)$ and $df/dg$ no longer have $\epsilon$ dependence and can be pulled out of the integral.
> > > First, this yields
> > > $$ \int d\epsilon \kappa_I(I+\epsilon^iL_i) f(g) = c f(g) \int d\epsilon \delta_\eta = c f(g) = W^0 f(g)$$
> > > Using $W^0 = c$, the second term is
> > > $$W^0 \left[\int d\epsilon \delta_\eta(I+\epsilon^j L_j) \epsilon^i\right] gL_i \cdot\frac{df}{dg} = W^0 \bar{\epsilon}^i L_i \cdot \frac{df}{dg}$$

---

### Official Review · Reviewer_DTit · 2021-07-16

**Rating:** 7
**Confidence:** 3

**Summary:**

The paper proposes a method to build equivariant neural networks by defining the layers in terms of the generators of the Lie group to which the NN is to be equivariant to. At the core lies the idea that any function (or conv kernel) with local support can be expanded in the Lie algebra basis via a Taylor approximation-like approach. Then, by observing that any function can be approximated with a basis of (shifted) local functions, it can be concluded that the presented approach in theory can approximate any group convolution. In addition, the authors the investigate the option to let go of the user-imposed constraint of being equivariant to a specific group, and instead let the neural network learn the generators (of the unknown group) in addition to the learnable weights that parametrize the layers. This is an open problem and the authors make promising steps in this direction.

Finally, the authors draw connections to physics, but I must admit, this part was too much for me and I did not get it, but possibly could have by investing more time in the paper.

**Ethical Concerns:**

None.

**Limitations And Societal Impact:**

Yes.

**Main Review:**

[overall impression]
Overall the paper presents very interesting ideas and has high theoretical value. Especially the Lie algebra approach in which convolution kernels are treated as a superposition of localized functions is appealing and leads to a new class of implementations of group convolutions. Additionally it allows to further explore the option to learn the symmetries that are relevant for a task. The unique approach gives new insights in the field of equivariant deep learning and that's where the value of the paper is. Furthermore, the paper is packed with important relevant insights, but at the same time I consider this it weakness as it makes it hard to distill the important parts. I do believe the paper can still significantly improve from more intuitive explanations; after rereading the paper and writing the review I am starting to see the brilliance of it but it does require quite some effort. So despite it being a paper that is hard to comprehend (could be just me), the vastness of sound theoretical insights makes the paper in my opinion worthwhile a publication at NeurIPS.

[readability/too much content]
As mentioned, my main concern is the density of the paper. In my opinion the paper is quite technical and there is simply too much information in it. It would have been nice if it were more focussed. As a reader I often struggled to grasp the general ideas as the emphasis is mostly on the technical details and I spend a lot of time on getting intuition or interpretation right. Providing layman's explanation/interpretation of the equations could improve readability of the paper (even for people already familiar to Lie group equivariant methods). More intuition probably comes at the cost of reducing the amount of information in the main paper. In my opinion, section 6 was a bit detached from the rest of the paper and was probably not necessary. I do appreciate such connections but it felt a bit too much (it could form the basis for a separate paper).

My detailed comments are as follows.

[discretization issues?]
Abstract, intro and Line 323: "... to encode many irreps or discretizing the group":
A key feature of the method seems to be that it doesn't require group discretization as opposed to other methods, but I am not sure if this is really the case. The definition of the L-conv layer has an obvious dependency of g on the left-hand side of the equation. Doesn't this mean one has to evaluate it on a grid on the group G? I think here I misunderstood something as it seems that in Equation 16 the dependency on g is lost. If I am indeed mistaken here, please use my ignorance as queue that more intuition about the equations need to be provided in the paper.

Regardless, the data is always discretized (see section 4 tensor notation) and so will the generators/left-invariant vector fields (or at least the regular representations/action of g on the vector representations of the data). So I don't understand the argument of not using a discretization as the proposed method still seems to be prone to discretization artifacts. If possible, try to address my concern regarding the discretization issue.

[Lie conv methods are involved?]
Line 32. "however, their approach is quite involved, ..." and line 39 "and this task is quite tedious". I disagree here, I think such Lie group based methods are in fact one of the easiest ways to build G-CNNs (especially compared to steerable G-CNNs) as it allows for a generic and modular code base in which only a few operations (group product, inverse and log) need to be defined after which implementations follow standard procedures. See e.g. the PoinConv method in Finzi et al. (already cited) or on a dense grid as in Bekkers et al. (missing cite, see ref below).

*ref*
Erik J Bekkers. B-spline cnns on lie groups. In International Conference on Learning Representations, 2019.

[group quotients?]
With respect to the lifting, i.e., replacing points x=g x0 with g. I think it is important to remark that g may not be unique (e.g. when x comes from a homogeneous space that is a quotient of the group, e.g., when x \in R^2 and G=SE(2)). It currently reads as if there is only one g for each x. If this is not the case, is this a problem?

[Equation 6]
Equation 6 is important but it is hard to follow what is precisely happening. Some intuition here may be helpful.
If I understand correctly the function f(g) is locally approximated with a first order Taylor expansion?
And because the kernel is localized the evaluation boils down to multiplying this approximation locally with W^0 as this like the average value of that the kernel is assumed to have and can locally be assumed to be constant?
Then d/dg is a bit abstract, this is "just" the differential of f, right? Which is a covector and multiplying it with g L_i filters out the i-th component?

A thing that confused me is that now all of a sudden the integration measure moves up-front in the integral whereas in eq 3 and 4 it is at the end. Is there a particular reason for this?

[Figure 2 is very insightful! ]

[Related work]
The approach taken in this paper turns out to be very related to two particular works. Firstly, also in Bekkers' work on B-Spline CNNs on Lie groups, convolution kernels are expanded in a basis of highly localized functions, namely B-splines. See e.g. Figure 3 of that paper. The result of group convolution with such kernels can then be obtained by expnding the kernels and to the regular g-conv, or by shifting the underlying function to the original and take the inner products with the basis functions, which is similar to the work presented in this paper.

Secondly, the work by Smets et al. on PDE based G-CNNs seems very related. In it, layers are defined by PDEs expressed in left-invariant vector fields, similar to this work. In particular, the convection term of their PDEs is computed by 1, constructing left-invariant vector fields from a given Lie algebra basis, 2 use them as differential operators (like in this paper equation 10) and 3, multiply them with corresponding weights (like in this paper). The viewpoint is however rather different (PDE vs local approximation viewpoint).

*ref*
Smets, B., Portegies, J., Bekkers, E., & Duits, R. (2020). PDE-based group equivariant convolutional neural networks. arXiv preprint arXiv:2001.09046.

[deep L-conv networks?]
It is a bit unclear how the experiment with learning the symmetries could generalize to deeper architectures. The current experiments only use one group equivariant layer (as 3 recurring L-convs). Are such layers also suitable for constructing deep architectures?
If so, would this mean that the generators need to be the same throughout all layers?

[practical details of learning the generators]
Line 199: "The L_i can be either inserted... or learned to discover symmetries" In the latter case, how is L_i parametrized? This is generally a very large and dense matrix, or am I mistaken? Is there any constraint put on the L_i matrices that are to be learned?




**Time Spent Reviewing:**

12

---

> ### Author Response · Authors · 2021-08-09
> **Part 1**
>
> > [overall impression]... superposition of localized functions... The unique approach gives new insights in the field of equivariant deep learning and that's where the value of the paper is.
>
> Thank you. We agree that the L-conv being a universal approximator for G-conv is the most important message of the paper.
>
> > ... paper is packed with important relevant insights, but at the same time I consider this it weakness as it makes it hard to distill the important parts... significantly improve from more intuitive explanations...
> [readability/too much content] my main concern is the density of the paper... quite technical and there is simply too much information... nice if... more focussed... struggled to grasp the general ideas...
> Providing layman's explanation/interpretation of the equations could improve readability...
>
> Very good point. We fully agree that more intuitive explanations would significantly improve the readability of the paper. We felt it was necessary to elaborate the technical details at the risk of making it hard to read. We will attempt to provide more intuition and better expplain the goal of each part.
>
> > More intuition probably comes at the cost of reducing the amount of information in the main paper. In my opinion, section 6 was a bit detached from the rest of the paper and was probably not necessary. I do appreciate such connections but it felt a bit too much (it could form the basis for a separate paper).
>
> We agree. We felt establishing the theory in depth was worth the risk. We also felt some concepts were too different from prior work to not discuss in detail: from the vector field $\hat{L}$ to how to implement it on discrete data (tensor notation) to how it connects with existing work in physics. Regarding section 6, it is true that it differs significantly from the literature on equivariance. But we realized that properties of equivariant cost functions are rarely discussed. This may be why connections with the extensive physics literature on equivariance had been missed. We felt the discussions at the end of the paper were a good place to show these connections. While we are working on a paper dedicated to the physics connection, we also wanted to share this with community to allow them to contribute.
>
> > [discretization issues?] Abstract, intro and Line 323: "... to encode many irreps or discretizing the group": A key feature of the method ...doesn't require group discretization... but I am not sure if this is really the case. ... it seems that in Equation 16 the dependency on g is lost.
>
> The dependency on $g$ is from the lift (e.g. $f(g)$ on an image is the value of colors on the pixel $x=gx_0$, where $x_0$ is the first pixel). If the data is on a discrete grid (like images) then there is a spatial discretization (tensor notation), but note that this does not mean the _group_ is discretized (see Appendix B.2) For example, images can be translated by a fracion of a pixel using Whittaker-Shannon interpolation (Appendix B.2 and C) or rotated by continuous angles. In eq. 16, $g$ becomes the index $\mu$. Recall that in the tensor notation, we have a discrete set of points $x_\mu$, which are lifted to $g_\mu\in G$ and $f(g)$, which is $f(g_\mu x_0)$ becomes
>
> $$ f(x_\mu) = f(g_\mu x_0) = x_0^T g_\mu^T \mathbf(f) \equiv \mathbf{f}_\mu $$
>
> which is just index $\mu$ of the vector $\mathbf{f}$. In the left part of eq. 16 we are not multiplying by $x_0^T g_\mu^T$ from the left and that is why it does not have the $\mu$ index. This means instead of looking at one index of the vector $\mathbf{f}$, we are looking at the whole vector (meaning instead of $f(g)$, we are looking at the function $f$ over the full domain). In the right part eq. 16, we are evaluating it at a specific $x_\mu$ and showing the indices explicitly. To be specific
>
> $$\hat{L}_\mu^\nu =[g_\mu L x_0]^\nu, \quad \mathbf{f}_\nu^c = x_0^T g_\nu^T \mathbf{f}^c $$
>
> We agree this is a difficult technicality to convey. The key is that $G$ can be continuous on discrete spaces. The tensor notation shows that on these spaces the whole vector field $\hat{L}$, encoding the flow of symmetry directions, becomes a matrix with components $[\hat{L}_i]_\mu^\nu$ which we can learn. This also reveals that L-conv can be implemented similar to GCN (as in eq. 16) and thus easy to work with.
>
>
> > Regardless, the data is always discretized (see section 4 tensor notation) ... proposed method still seems to be prone to discretization artifacts.
>
> Note that the data does not always need to be discrete. For instance, data on the dynamics of objects can be continuous. As long as partial derivatives $\partial_\alpha f$ (or velocity vectors) are provided, one can apply the continuous version of L-conv to the data. When the data is discrete, the apparent symmetry group may be slightly different from the symmetry group of the underlying continuous space (see Appendix B.2).
> However, in many cases the symmetry of the discrete space can be continuous and isomorphic to the continuous version. As discussed in the paper, the Whittaker-Shannon theorem allows us to define continuous translations and rotations isomorphic to their continuous counterparts. So the discretization in the tensor notation is not discretizing the group.
>
> > [Lie conv methods are involved?] ... one of the easiest ways to build G-CNNs ... as it allows for a generic and modular code base in which only a few operations (group product, inverse and log) need to be defined ... Bekkers et al. (missing cite, see ref below).
>
> Thank you for the reference. We will add a discussion of these. LieConv is certainly a great work. But as you also mention, the log, product and inverse need to implemented for each group and each representation separately. In comparison, our method is similar to GCN. Even when the group is known, we only need to encode $L_i$ and no group specific functions.
>
> > [group quotients?] ...g may not be unique... is this a problem?
>
> That is correct, but the non-uniqueness should be only due to the stabilizer, meaning if $gx_0= g'x_0$ then $hx_0=x_0$ where $h=g^{-1}g'$ form a subgroup $h\in H\subseteq G$, the stabilizer of the lift. We left out this discussion as it has been extensively addressed in Kondor and Trivedi. We only briefly mention it for the loss function, in Appendix A.3. Note that the stabilizer impacts the lift $g$, but even if $L_i$ include the Lie algebra of $H$, since their action on the origin vanishes, $L_i x_0=0$, including or excluding shouldn't matter. This probably also means that the stabilizer cannot be learned by L-conv.  Thus we are implicitly assuming we are working with $G/H$, where $H$ is the stabilizer of the origin. We can add a discussion if you think it is necessary.
>
> > [Equation 6] Equation 6 is important but it is hard to follow what is precisely happening. ...
>
> We agree. We will reorganize eq. 6 and add more details from eq. 25 to it to clarify. Yes, we are Taylor expanding $f(gv_\epsilon)= f(g(I+\epsilon^i L_i))$ to first order n $\epsilon$. The $W^0$ arises from integrating the $\epsilon$-independent term $f(g)$, meaning
> $$ c \int d\epsilon \delta_\eta(I+\epsilon^i L_i) f(g) = W^0 f(g).$$
> The kernel being localized in a small $\eta$ neighborhood is what allows us to neglect higher-order terms in the Taylor expansion, but it doesn't play a role in defining $W^0$ (as long as we can Taylor expand $f(gv)$ in $v$ the first term will be a constant $W^0$ times $f(g)$).
>
>
> > Then d/dg is a bit abstract, this is "just" the differential of f, right? ...
>
> Yes, just the differential, as eq. 25 clarifies. We are expanding $f(g+\epsilon^i gL_i)$ around $g$. The more mathematically rigorous expression would be
>
> $$ f(g+\epsilon^i gL_i) = f(g) + [\epsilon^i gL_i]^\alpha_\beta \left.\frac{d f(u)}{du^\alpha_\beta}\right|_{u\to g} + O(|\epsilon|^2)$$
>
> > A thing that confused me is that now all of a sudden the integration measure moves up-front in the integral whereas in eq 3 and 4 it is at the end. Is there a particular reason for this?
>
> Maybe the $W^0$ is confusing. Note that in the first line of eq. 6 _everything_ except $O(\eta^2)$ is inside the integral. The second term is
> $$c\int d\epsilon \left(\delta_\eta(I+\epsilon^j L_j) \epsilon^iL_i \cdot\frac{df}{dg} \right)= W^i g L_i \cdot\frac{df}{dg} \equiv W^0 \bar{\epsilon}^i L_i \cdot \frac{df}{dg}$$
>
> where we decompose $W^i$ as $W^0\bar{\epsilon}$. The reason was to clarify that the magnitude of $W^i$ is smaller than $W^0$ by order $\eta$. Also, when $\delta_\eta$ is normalized, $W^0=c$ (note the missing $c$ in the definition of $\bar{epsilon}$ in eq. 7). We can clarify this in the main text.
>
> > [Related work] The approach taken in this paper turns out to be very related to two particular works. Firstly, also in Bekkers' work on B-Spline CNNs on Lie groups, convolution kernels are expanded in a basis of highly localized functions, namely B-splines. ...
>
> Thank you for bringing this to our attention. Yes, the idea of our Figure 2 is very similar to their expansion. We are only extending that by saying the localized basis itself can be implemented as recurrant or multi-layer L-conv.
>
> > Secondly, the work by Smets et al. on PDE based G-CNNs...
>
>
> Interesting, thank you. Indeed eq. 17 in this paper is very similar to our $\hat{L}$ and eq. 19 the same as our examples for translation $T_1$ and rotation $SO(2)$. We will cite this paper and study it more closely. In a way, L-conv is the differential of a G-conv layer and passing through an L-conv layer is a PDE for moving the input along the flow of symmetry.

---

> > ### Comment · Reviewer_DTit · 2021-08-20
> > **Response**
> >
> > Thank you for the elaborate answers! This has been very helpful, although I must still admit I am still a bit struggling with the discretization statements such as :
> >
> > "Existing equivariant neural networks for continuous groups require discretization or group representations"
> > > So does this method if I understood correctly. Namely, if the data is a discretized function (as is in the data considered in this paper), the output of the layer still has to be sampled on a grid (indexed with mu in (16)) in order to be able to compute with it (unless perhaps working with continuous data on which you can apply the generators as differential operators). Existing equivariant methods have the same "issue", they are build with tools for continuous groups and can also act on discrete data (either by interpolation or basis/Fourier expansion methods). In this work the Lie algebra generator is discretized with a matrix (whose size scales rapidly) with the resolution of the discretization.
> >
> > "Our model, the Lie algebra convolutional network (L-conv) can learn potential symmetries and does not require discretization of the group"
> > > After the explanations I can see that a discretization of the group is not always necessary, but I have the feeling that still you need to learn a finite set of generators in order to learn interesting things. Like with group convolutions (section 3.1), you are learning a finite set of left-translations that can be linearly combined to form a group convolution kernel. One can argue that eventually one still suffers from discretization, either because the data is discrete or because you want to learning sufficiently expressive networks by considering more generators (or both).
> >
> > So, even though I found the paper very intriguing, I am still not sure why so much emphasis is put on avoiding discretization. I think the point can still somehow be made, but I didn't get it at first and I still struggle to get it. Additional explanation or intuition will be helpful. In particular, I think it would help to make explicit in what spaces the inputs and outputs live.

---

> > > ### Author Response · Authors · 2021-08-21
> > > **Difference between discretization in our method and other approaches**
> > >
> > > > [require discretization] So does this method if I understood correctly. Namely, if the data is a discretized function (as is in the data considered in this paper), the output of the layer still has to be sampled on a grid (indexed with mu in (16))
> > >
> > > We want to emphasize that the spatial discretization (tensor notation) is also present in other methods, for instance when dealing with images. In addition to this, many other methods _discretize the group manifold_  by sampling a finite number of _group elements_ $g_1,\dots g_m \in G$. We do not discretize the group. We rely on the fact that a finite number of $L_i$ can generate a continuous group. For example, the $\hat{L}$ learned in figure 3 for $SO(2)$ can be used to generate rotations with arbitrary, continuous angles. Similarly, the translation generator in line 228 and Figure 6 generates a continuous translation group. So, note that even though it acts on a discretized 1D space with $d$ pixels, the group generated by $\hat{L}$ in Figure 6 is the continuous 1D translation, capable of shifting 1D signals by a fraction of a pixel (also see Rao 1999).
> > >
> > > > After the explanations I can see that a discretization of the group is not always necessary, but I have the feeling that still you need to learn a finite set of generators in order to learn interesting things...
> > >
> > > Note that we only need infinitesimal generators $L_i$. We believe the experiment in Figure 3 runs counter to your argument. In Fig. 3 we use recurrence to learn only a single $L$ which can generate the continuous group $SO(2)$. Using the procedure in sec. 3.1 we can use this single $L$ to approximate any kernel on $SO(2)$. So in order for L-conv to be expressive it only needs to learn an $L_i$ which form a basis for the Lie algebra. The number of $L_i$ needed is generally small, equal to the dimension of the Lie algebra of the symmetry group. Thus learning more $L_i$ than required would not translate to higher expressivity.
> > > And note that discretization of space did not prevent us from having a continuous $SO(2)$ symmetry. It just resulted in non-trivial $L_i$ as in Fig. 7 and Fig. 6 for continuous translation.
> > >
> > >
> > > > So, even though I found the paper very intriguing, I am still not sure why so much emphasis is put on avoiding discretization. I think the point can still somehow be made, but I didn't get it at first and I still struggle to get it. Additional explanation or intuition will be helpful. In particular, I think it would help to make explicit in what spaces the inputs and outputs live.
> > >
> > > Avoiding discretization is not the main goal, but it is an added benefit of using the Lie algebra. A good intuition could be this: $L_i$ are like basis vectors. To encode an arbitrary vector in, say 2D  we only need 2 coefficients for $\hat{x},\hat{y}$ basis vectors. Discretizing the group, on the other hand, is like sampling $n$ grid points in 2D. To encode an arbitrary vector in the discretized version one finds which of the $n$ points is closest to the given vector. So the real difference is we are learning $\theta^i$ in $g = \exp[\theta^iL_i]$, whereas G-conv is learning  $c_k$ in $ g = \sum_{k=1}^n c_k g_k$ to find which sample $g_k\in G$ is closest to a given $g$. By learning $\theta_i$ instead of $c_k$ we remove the group sampling step, which is the harder part to implement. Additionally, learning the $\theta^i$ using MLP allows for much more flexibility. For example, if the data only has small rotations but $SO(2)$ is discretized by sampling $g_k$ rotations with large angle increments, G-conv won't be able to learn the small rotations, but MLP learning $\theta^i$ should identify and learn the right range for the rotations. To increase accuracy in the discretized case we need to sample more points, whereas in our case accuracy depends on the accuracy of the two coefficients and how accurately we encode the basis vectors (i.e., $L_i$).
> > >
> > > Regarding the input and output spaces of L-conv: in the tensor notation, the input $\mathbf{f}$ and the L-conv output $Q[\mathbf{f}]$ are in
> > > $$\mathbf{f}\in \mathcal{S}\otimes \mathcal{F}=\mathbb{R}^d \otimes \mathbb{R}^{m_{in}}, \quad Q[\mathbf{f}]\in \mathcal{S}\otimes \mathcal{F'}=\mathbb{R}^d \otimes \mathbb{R}^{m_{out}}$$
> > >
> > > Both are distributions over the same base space $\mathcal{S}$. Only the number of feature dimensions differ. For example, an RGB image passing through L-conv has three input feature dimensions and will have $m_{out}$ output channels when the L-conv weight $W^0$ is $3\times m_{out}$.
> > > The full size of the input image $\mathbf{f}$ in tensor notation is $d\times 3$ where $d=H\times W$  is the number of pixels. The output of L-conv $Q[\mathbf{f}]$ has dimensions $d \times m_{out}$.

---

> > > > ### Comment · Reviewer_DTit · 2021-08-23
> > > > **Thank you for the discussion.**
> > > >
> > > > Thank you for the added explanation!
> > > >
> > > > Firstly, my apologies for not being able to accurately formulate my questions. It has already been clear to me that the group itself does not need to be discretized as I understand how generators work and can act on discrete data, I think my problem, however, is that I have difficulties in thinking about how to construct useful architectures without lifting the data to the group. I’m trying to relate the work to standard G-CNNs, which perhaps I shouldn’t, as the proposed method requires a different way of thinking about building architectures. I do not expect a detailed answer, but feel free to do so. The following is still on my mind.
> > > >
> > > > My issue is that  I get stuck when trying to figure out how to build e.g. SE(2) equivariant architectures for 2D images. To me this is most naturally done by lifting the images to (discretized) functions on SE(2) for which we have 3 natural generators that generate shifts relative to the local positions/orientations and I can clearly see how to link it to g-convs. However, then we are back to the discretization of the group which the proposed method want to avoid. When we do not lift (so we only have the spatial pixel locations), then there is ambiguity in the origin around which to rotate, which does not commute with translations, and I do not understand in what sense the method can relate to group convolutions (I currently think it can’t). I lack intuition on how L-conv would achieve equivariance to the full group SE(2).
> > > >
> > > > Regarding the “only” a single generator is sufficient argument: The approximation of g-convs relies on expanding a kernel as a sum of shifted localized basis functions. Then each shift u_k is obtained with a single generator that is iterated (or even multiple ones (cf. line 137-138). Then this needs to be done for multiple shifts and thus one needs more that 1 generator to construct the output of a group convolution. So, even though a single generator can generate the full group, trying to learn a group convolution operator requires many (at least the number of basis elements K). Alternatively, since it is likely that the kernel is sampled at points along the same orbit (which is surely the case for 1D groups), one could reuse the generator and use something like skip connections to “store” values sampled at shifts along the orbit. I can vaguely see how this can be done with the proposed architecture for the rotation case SO(2), where deep L-Conv networks could in principle be able to combine the results of multiple input rotations. I cannot see how this works for SE(2) equivariant methods applied to 2D images, unless the data is lifted to the group.
> > > >
> > > > Your answer above regarding learning $\theta_i$ vs the $c_k$ touches upon this topic and has been insightful. However, I think it assumes that there is only a single group element relevant which needs to be learned. This seems limiting from the G-conv point of view where one looks for patterns for relative transformations (/angles).
> > > >
> > > > A final remark on *learning symmetries*, it is clear that the method can adapt to problems with unknown symmetries in them. It is however, unclear if this should be interpreted as a method for actually *discovering* symmetries as I am not sure how easy it will be to "read-off" the symmetries contained in the learned L_i. How should I interpret the title "Automatic Symmetry Discovery ..."?
> > > >
> > > > **[In conclusion]**
> > > >
> > > > In the end it boils down again to presentation and adding intuition to accompany the math. As mentioned before, I found the paper lacks intuition regarding such aspects and misses clear handles for mathematically inclined engineers (such as I) on how to build NNs with them. Though, as already confirmed by the authors, this will be improved in a revision. The discussion did give me great insights which reinforces my belief in the relevance of the paper.

---

> > > > > ### Author Response · Authors · 2021-08-30
> > > > > **Clarification about generators and architecture**
> > > > >
> > > > > > I think my problem, however, is that I have difficulties in thinking about how to construct useful architectures without lifting the data to the group.
> > > > > My issue is that I get stuck when trying to figure out how to build e.g. SE(2) equivariant architectures for 2D images. To me this is most naturally done by lifting the images to (discretized) functions on SE(2) for which we have 3 natural generators that generate shifts relative to the local positions/orientations and I can clearly see how to link it to g-convs. However, then we are back to the discretization of the group which the proposed method want to avoid.
> > > > >
> > > > >
> > > > > Thank you for the clarification on this important point. To make sure we understand the issue: you are considering an image as a discrete function over the group, meaning an image is $f:\mathcal{S}\to G$, where $\mathcal{S}=\{x_0,\dots x_d\}$ are the pixel locations.
> > > > > This part is the same as our tensor notation, we represent $f$ as a vector $\mathbf{f}$ with components $\mathbf{f}_\mu= f(x_\mu)$. So, when $x_\mu = g_\mu x_0$ are lifted to the group elements $g_\mu$, the image $\mathbf{f}$ is also lifted to the group. The origin $x_0$ is also chosen the same way as in G-conv.
> > > > >
> > > > > Then, for $SE(3)$, the three generators are shifting and mixing $f(x_\mu)= \mathbf{f}_\mu$ among the neighbors of each $x_\mu$. This is again the same in the tensor notation, where $\hat{L}_i$ are $d\times d$ matrices acting on $\mathbf{f}$.
> > > > > The point is $\hat{L}_i$ can be constructed such that they can create infinitesimal and continuous elements of $SE(3)$ not just a set of discrete ones.
> > > > >
> > > > > The confusion may be about what the infinitesimal shift generator is.
> > > > > One might think that the infinitesimal generator of shift to the right in an image is the circulant matrix shifting by one pixel to the right, but this not the case.
> > > > > The correct infinitesimal shift generator is the one in line 228, which is more complex. This generator does not discretize the group and arbitrary continuous shifts can be generated with it. Note continuous shifts in a discrete image are possible. For example, a shift by half a pixel to the right would have two neighboring pixels at half the intensity.
> > > > > Is this the type of discretization you are concerned about and does this answer your question?
> > > > >
> > > > >
> > > > >
> > > > >
> > > > > > Regarding the “only” a single generator is sufficient argument: The approximation of g-convs relies on expanding a kernel as a sum of shifted localized basis functions. Then each shift u_k is obtained with a single generator that is iterated (or even multiple ones (cf. line 137-138). Then this needs to be done for multiple shifts and thus one needs more that 1 generator to construct the output of a group convolution.
> > > > > So, even though a single generator can generate the full group, trying to learn a group convolution operator requires many (at least the number of basis elements K).
> > > > >
> > > > > To clarify, by "each shift u_k is obtained with a single generator that is iterated (or even multiple ones (cf. line 137-138)" you are not referring to $v_{\epsilon_a}$ as generators, correct? The infinitesimal generators $L_i$ are the same for shifting all $u_k$. For each $u_k$ we need a set of $v_{\epsilon_{ak}}= I+\bar{\epsilon}\_{ak}^i L_i$ such that $u_k=\prod_{a=1}^p v_{\epsilon_{ak}} $, but the $L_i$ in all of them are the same.
> > > > > Here we included an explicit index $k$ to clarify that $\bar{\epsilon}\_{ak}$ encodes one of the $u_k$.
> > > > > Since $v_{\epsilon_{ak}}$ has the structure of an L-conv layer (with $W^0=1$), the product $u_k=\prod_{a=1}^p v_{\epsilon_{ak}} $ is a multilayer L-conv with learnable parameters $\bar{\epsilon}\_{ak}^i$ and a set of shared, learnable infinitesimal generators $L_i$. So we don't need more than one set of infinitesimal generators $L_i$.
> > > > > To learn the $u_k$ with $k=1,\dots n_k$, we can take two approaches:
> > > > >
> > > > > __Multi-layer L-conv approach:__
> > > > > We can make a $p$ layer L-conv, with shared $L_i$, where the parameters of layer $a$ for each $L_i$ are length $n_k$ vectors $\theta_a^i$ with components $[\theta_a^i]_k=\bar{\epsilon}\_{ak}^i$.
> > > > > The number $n_k$ of $k$ is a hyperparameter, determining how many $u_k$ we can learn.
> > > > >
> > > > > __Recurrent L-conv approach:__
> > > > > Another approach is to stack the $p$ length $n_k$ parameter vectors $\theta_a^i$ for all $a$ into one $p\times n_k$ matrix $\theta^i$.
> > > > > Then, instead of having $p$ layers of L-conv, we make a $p$-times recurrent L-conv with parameter matrix $\theta^i$. In practice, it is easier to implement the weight matrix as a square matrix so that the recurrence can be done easily. To do so, we make $\theta^i$ an $m\times m$ matrix where $m= \max \\{p,n_k\\}$.
> > > > > This is the approach we took in the experiment of Figure 3.
> > > > > As detailed in SI C.3, we learn a $10 \times 10$ matrix $\theta$ for $\bar{\epsilon}\_{ak}$ using a $p=3$ times recurrent architecture.
> > > > > We have a pair of images rotated relative to each other.
> > > > > The recurrent L-conv learns to rotate the first image by different angles appearing in the data.
> > > > > These different rotated outputs are then dotted into the second image and final MLP learns to estimate the angle from the result of the dot products.
> > > > >
> > > > >
> > > > > > Alternatively, since it is likely that the kernel is sampled at points along the same orbit (which is surely the case for 1D groups), one could reuse the generator and use something like skip connections to “store” values sampled at shifts along the orbit. I can vaguely see how this can be done with the proposed architecture for the rotation case SO(2), where deep L-Conv networks could in principle be able to combine the results of multiple input rotations. I cannot see how this works for SE(2) equivariant methods applied to 2D images, unless the data is lifted to the group.
> > > > >
> > > > > The structure of L-conv $W^0(I+\epsilon^iL_i)$ naturally has a skip connection due to $I$, so you are correct about that.
> > > > > This enables a $p$ layer L-conv to access all order $p$ products $L_iL_j\dots L_k$.
> > > > > However, note that a $p$ layer L-conv is not only $g^p$ for some $g\in G$ and so not just the orbit of one group element.
> > > > > Like above, a $p$ layer L-conv has a structure $\prod_a (I+\epsilon_a^iL_i)$.
> > > > > Thus a $p$ layer L-conv is like taking $p$ steps on the group manifold, where step $a$ has step size $\epsilon_a^i$ in the direction of infinitesimal generator $L_i$ (pushed forward).
> > > > > So each step can be in an arbitrary direction on a multi-dimensional group manifold, not necessarily the orbit of one group element.
> > > > > For example, for $SE(3)$, the first L-conv layer may move in the $x$ direction and the second layer in the $y$ direction.
> > > > > As discussed above, by letting each L-conv layer have a matrix of parameters with components $\epsilon_{ak}^i$ a multi-layer L-conv can reach many different $u_k$, including when there are multiple $L_i$.
> > > > >
> > > > >
> > > > > >Your answer above regarding learning $\theta_i$ vs $c_k$ the touches upon this topic and has been insightful. However, I think it assumes that there is only a single group element relevant which needs to be learned. This seems limiting from the G-conv point of view where one looks for patterns for relative transformations (/angles).
> > > > >
> > > > > Actually, $\theta^i$ can be a matrix, as we discussed above. As in the Fig 3 example, L-conv is not learning $\theta$ for a single group element, but rather a set of $\theta^i_k$ which can be combined to make a large set of group elements, using recurrence or multi-layer architecture.
> > > > >
> > > > > > A final remark on learning symmetries, it is clear that the method can adapt to problems with unknown symmetries in them. It is however, unclear if this should be interpreted as a method for actually discovering symmetries as I am not sure how easy it will be to "read-off" the symmetries contained in the learned L_i. How should I interpret the title "Automatic Symmetry Discovery ..."?
> > > > >
> > > > > In terms of reading off the symmetry generators, it simply means read the value of $L_i$ parameters in the L-conv layers.
> > > > > For exmaple, both in Figure 3 and for the experiemnts in appendix D we can simply read the value of the $L_i$ in the L-conv layer, but perhaps you mean understanding what those values mean can be difficult?
> > > > > If this is your question, we can provide suggestions as to how to interpret them.
> > > > > For exmaple, the $L$ learned in Figure 3 can be used to create a vector field showing the flow of the rotation generator in 2D.
> > > > > To do this, we read row $m$ of $L$ and index the entries by which pixel they correspond to.
> > > > > The elements close to the diagonal in $L$ encode mixing with nearby pixels in the x direction of the image and lines parallel to the main diagonal in $L$ represent mixing with nearby pixels in the $y$ direction.
> > > > > These values can be seen as a weighted sum over nearby pixels for each row $m$, defining a vector for the direction of the flow of $\hat{L}$ at pixel $m$.
> > > > > By "Automatic symmetry discovery" we mean that if continuous symmetries exist in a problem, the generators $L_i$ of the symmetry are automatically learned during training.
> > > > >
> > > > > > [In conclusion] In the end it boils down again to presentation and adding intuition to accompany the math. As mentioned before, I found the paper lacks intuition regarding such aspects and misses clear handles for mathematically inclined engineers (such as I) on how to build NNs with them. Though, as already confirmed by the authors, this will be improved in a revision. The discussion did give me great insights which reinforces my belief in the relevance of the paper.
> > > > >
> > > > > Thank you. This is an important suggestion. We will strive to add more intuitive discussions.

---

> ### Author Response · Authors · 2021-08-09
> **Part 2**
>
>
> > [deep L-conv networks?] It is a bit unclear how the experiment with learning the symmetries could generalize to deeper architectures. The current experiments only use one group equivariant layer (as 3 recurring L-convs). Are such layers also suitable for constructing deep architectures? If so, would this mean that the generators need to be the same throughout all layers?
>
> This is a very good question. Three approaches come to our mind. First is sharing $L_i$ across layers, either in a recurrent architecture like Fig. 3, or only sharing $L_i$, while allowing $W^0$ and $\bar{\epsilon}$ to differ in layers. In this approach the assumption is the layers are encoding the same symmetry, but the weights are learning general functions. Note that, as discussed in Appendix A "Extended equivariance for L-conv" (eq. 29 and 30), $W^0$ can become any arbitrary DNN without affecting the equivariance.
> The second approach is learning different $L$ for each layer of L-conv. In our 2-layer experiments, this did not yield significant improvements over shared $L_i$.
> A third approach when spatial pooling is included. Each pooling coarsens the data in a new scale (losing finer details) and the symmetries at the new scale are not guaranteed to be the same as the finer scale. Therefore, it is better to learn new $L_i$ after each coarsening layer. Nevertheless, it is possible to reuse $L_i$ even after coarsening. The key is to use eq. 17. In this eq. 17 $\hat{{l}}_i$ is the action of the Lie algebra on the neighbors of identity. After Coarsening, we can keep $\hat{{l}}_i$ the same while changing $\mathbf{B}$, which encodes the grid of the discretized space, and the lift $g_\mu$.
>
>
> > [practical details of learning the generators] Line 199: "The L_i can be either inserted... or learned to discover symmetries" In the latter case, how is L_i parametrized? This is generally a very large and dense matrix, or am I mistaken? Is there any constraint put on the L_i matrices that are to be learned?
>
> Correct. The simplistic approach is to encode it as a full matrix, as was done in Fig. 3. The more accurate approach is to encode it using eq. 17, which means we learn a small length $k$ vector $\hat{{l}}_i$ ($k$ being the number of neighbors of each node mixed through an infinitesimal group action) and a shared $d\times k$ lift matrix $g_\mu$. The incidence matrix $\mathbf{B}$ encoding the topology is generally known, but in problems like the scrambled-rotated images, even $\mathbf{B}$ is unknown. In that case we can learn $g_\mu\mathbf{B}$ as one matrix. In our experiments in appendix D we chose the simpler approach of encoding $L_i$ as low-rank matrices (see Appendix D, "Test Model Architectures").

---

### Official Review · Reviewer_vpfc · 2021-07-21

**Rating:** 6
**Confidence:** 3

**Summary:**

They propose a model for approximating invariant functions for Lie groups, the Lie algebra convolutional network (L-conv). While convolutional networks for ordinary groups require some kind of discretization, their model does not require it. Also, Lie algebra generators in our L-conv can be learned to automatically discover symmetries.

**Ethical Concerns:**

No.

**Limitations And Societal Impact:**

No.

**Main Review:**

This was a difficult paper to judge. First of all, I can assure that the mathematical instrumentation of their method is solid.
LieConv. and other group convolutions have the disadvantage that the integral over the Haar measure needs to be approximated by a discrete approximation when implemented, which prevents it from being an equivariant function in the strict sense. In their method, the convolution for the group is approximated using an equivariant function, and the model gives a strictly equivariant function.If you look at Theorem 1, you will also see that their model is an universal approximator for functions written in group convolution. If it is an universal approximator for equivariant functions, it should be clearly stated.
I considered the experimental part of this paper to be weak. Although it has the strength of giving an exact invariant function, it is unclear in the current paper whether the previous methods or theirs are stronger for the actual invariant task. This is because their model also has a part that approximates the convolution, and it is not clear at this stage whether the approximation accuracy or the approximation accuracy by discretization is better.
It would be good to have some comparison experiments with Finzi's model, for example, to clarify this part.
There are no major problems with the structure of the paper.
Here are some typos I found.
105 expand
197 L-conv equation 58

**Time Spent Reviewing:**

20hours

---

> ### Author Response · Authors · 2021-08-06
> **Universal approximation and experiments**
>
> Thank you for your comments.
>
> > If it is an universal approximator for equivariant functions, it should be clearly stated.
>
> This is a good suggestion. As you stated from Theorem 1, L-conv is a universal approximator for group convolution, which represents linear equivariant functions. For general, nonlinear equivariant functions, under some hypotheses, it should be a quick corollary of our result that L-conv has universal approximation for equivariant functions.
> In particular, universal approximation theorems for equivariant neural networks have been proven under certain hypothesis (for certain classes of groups and certain types of functions):
> https://arxiv.org/abs/2010.02449 universal approximation for rotation on point clouds
> https://arxiv.org/abs/1903.01939 universal functional approximation for finite groups
> https://arxiv.org/abs/1901.09342 for invariant functions
> By https://arxiv.org/abs/1802.03690, under some hypothesis (compact G, transitive action) G-conv has greater or equal expressive power as any class of equivariant neural networks such as those considered in these papers. Thus under these assumptions, one may show that since L-conv approximates G-conv which is as expressive as classes which are proven to be universal approximators, L-conv is a universal approximator.
>
> > Although it has the strength of giving an exact invariant function, it is unclear in the current paper whether the previous methods or theirs are stronger for the actual invariant task. This is because their model also has a part that approximates the convolution, and it is not clear at this stage whether the approximation accuracy or the approximation accuracy by discretization is better.
>
> Correct. We would like to note that L-conv is not the best solution for all equivariant tasks. As Fig.12 (appendix D) shows, on the "Default" image datasets CNN outperforms L-conv while using fewer parameters. The message is that, if we have good inductive bias (e.g. an efficient implementation of a group equivariant model), that model can outperform L-conv, which is trying to learn the symmetry $L_i$ and solve the task at the same time. Where L-conv is beneficial is tasks where we have no knowledge of the possible symmetries.
>
> > It would be good to have some comparison experiments with Finzi's model, for example, to clarify this part.
>
> We already experimented with the standard tasks in Appendix D and provided a very narrow experiment necessitated learning the correct $L$. With further experiments (Supp, Figs. 11 and 13) we showed that even with less parameters L-conv was able to beat CNN in rotated and scrambled images.
> Our method cannot outperform baselines with known symmetry inductive biases. Rotated & scrambled MNIST is a good benchmark for methods which can exploit discovered unknown symmetry. This presents a data distribution which definitively has symmetry but which does not match the translational or rotational inductive biases of CNN or E(2)-CNN and cannot be easily programmed as an inductive bias into LieConv.  Real-world data which might match this could include radar scattering data or xray diffraction data which has rotational and translational symmetries within the underlying physics but are difficult to appreciate in their usually presented form.
>
> If you would like to see some other more specific experiment, please let us know.

---

> > ### Comment · Reviewer_vpfc · 2021-08-22
> > **Thanks for the reply.**
> >
> > Thanks for the reply.
> >
> > Regarding the universal approximation
> > I understand that it is a difficult problem to formulate for unknown groups. Your claim is that "a G-equivariant map that can be approximated by a composition of G-equivariant layers can be approximated by L-conv. I think some readers may want to know if there is universal approximation property, so I suggest you add what you wrote in your reply to the paper.
> >
> > About the experiment
> > I apologize that I missed the experiment on rotated and scrambled MNIST in the appendix.
> > This experiment helped me to understand the significance of some of your results.  I think that rotated and scrambled MNIST is an indisputable subject in terms of unknown group actions, but are there any more real-world unknown group actions?
> > Also, as for the point about not getting better results than the case of known group actions,  I am convinced that the bias is different.

---

> > > ### Author Response · Authors · 2021-08-29
> > > **Thank you for the sugestion**
> > >
> > > > Regarding the universal approximation ... I suggest you add what you wrote in your reply to the paper.
> > >
> > > Thank you, this is  a great suggestion. We are claiming that L-conv is a universal approximator for any G-conv, so we should make this explicit.
> > >
> > > > About the experiment... rotated and scrambled MNIST is an indisputable subject in terms of unknown group actions, but are there any more real-world unknown group actions?
> > >
> > > Yes, as we mentioned, radar or x-ray diffraction, which are transformed versions of spatial data, will have a different representation of spatial symmetries encoded in them and L-conv can help uncover such symmetries. In other cases, the data may have a symmetry which we did not suspect. For example, Graphene turned out to have an unexpected Lorentz symmetry.
> > >
> > > Similarly, in many dynamical systems, we know the dynamics has symplectic group as its symmetry but finding the correct basis where these symmetries become manifest is nonlinear and difficult. By learning the vector fields $\hat{L}_i$ encoding the flow of such symmetries, L-conv can potentially discover such symmetries of dynamics. L-conv could also be applied to latent space representations and find potential symmetries in the loss landscape. The conventional approach of encoding familiar symmetries into the architecture may miss such symmetries.

---

### Official Review · Reviewer_2v7e · 2021-07-25

**Rating:** 7
**Confidence:** 4

**Summary:**

In the manuscript, the authors propose a method to estimate continuous symmetry of data through the estimation of Lie rings, rather than the Lie groups themselves.
To the best of my knowledge, there has never been a method for nonparametric estimation of Lie rings with linear representations, and I believe that the method is very novel and significant in the study of equivariant neural networks.

**Limitations And Societal Impact:**

I described one of my concerns about the limitation of the method as major comment 1 on the Main review.
In addition to this, I point out that there is no discussion on the limitation of the method by restricting the Lie ring to realization by linear transformations. If possible, please discuss what kind of symmetries cannot be handled by the proposed method.


I do not think there is a negative societal impact.

**Main Review:**

As mentioned above, I believe that the method which is described in this manuscript has both significant novelty and importance that should be urgently shared with researchers in the same field.
Although I acknowledge the novelty of the proposed method, the current manuscript is unclear what is the purpose of the research and the novelty of the whole research resulting from the development of the method. I think that the paper should be accepted after this ambiguity has been fixed.
In the following, I pointed out the unclear points according to the novelty or research purpose pointed out by the authors.


1. We propose the Lie algebra convolutional network (L-conv), a building block for constructing group equivariant architectures.
>The authors state that the estimation error of the Lie group by the proposed method is $O(η^{p+1})$.
In the estimation of Lie rings, only information in the neighborhood of a point is used.
Therefore, the estimation error of the Lie group by the proposed method may be much larger than that of the direct estimation of the Lie group due to noise in the data, training error of DNN[1], or the large curvature of the manifold formed by the Lie group.
In this case, the accuracy of the task or the symmetry estimation will be lower than the previous studies that introduce the constraint of the Lie group or estimate the Lie group directly.
Can you provide any discussion or numerical experiments that show that this does not occur?


2. We show that multi-layer L-conv can approximate group convolutional layers, including CNN, and find graph convolutional networks to be a special case of L-conv.
>This is a very interesting discussion, but the meaning of this discussion in this research is unclear.
What advantage does this property give to L-conv as a network model of deep neural networks?
If the advantage includes higher performance compared to CNNs and GCNs, then the numerical results in the supplemental material D should be mentioned here as well.


3. The Lie algebra generators in our L-conv can be learned to automatically discover symmetries.
>As far as I know, there are methods that also learn Lie groups [2] or estimate Lie-ring from trained DNNs [1].
What is the advantage of the proposed method?


4. We find that MSE loss function for L-conv generalizes important Lagrangians in physics.
> I could not understand why "The MSE is given by $I$".
Are you making the assumption that the error is given in quadratic form?
If so, can $I$ represent any quadratic-form likelihood function with symmetries?


5. We find generalization error and equivariance in the loss landscape can be expressed as Euler-Lagrange equations and Noether currents, transferring ideas from physics to ML.
> About the discussion about this novelty (from Line 283 to 296), it seems to be obvious that if the Lagrangian satisfies the principle of least action (Robustness to the perturbation $\delta \phi$), then it satisfies Hamilton's principle (variational principle).
Therefore, it could not seem to be an element in itself to consist the novelty of the manuscript.
About the discussion about this novelty (from Line 297 to 306), the discussion could not seem to be an element in itself to consist the novelty of the manuscript either since the discussion only contains a description of the basic knowledge of analytical mechanics.
In order to increase the importance of the discussion, I would like to see a more detailed description of how the obtained knowledge can be used in concrete ways.
Specifically, I did not feel the following statements were concrete enough.
(Line 304) "It would be interesting to see if these conserved currents can be used in practice as an alternative way for identifying or discovering symmetries. "

6. There are the other major comments about "7 Discussion" as described below:
Authors have pointed out the following as useful aspects of the findings of this research.
(Line 325)"But perhaps the most interesting discovery here is the connection made with symmetry encoding models in physics. This is significant for two reasons. "
However, I could not determine if the two points presented have significance for the following reasons, respectively.

6-1. About (Line 326) "One is that methods for dealing with symmetry used in physics acquire important meanings in ML, as our discussion on the loss function, generalization and conservation laws show. "
>It is true that paragraphs such as Line 307 discuss the correspondence between physics and machine learning, but it does not specifically describe the prospects of how they can be connected to the development of machine learning algorithms.
Therefore, I can not understand there is any significance in this finding.


6-2. About "The second reason is for scientific machine learning. Physicists usually use the simplest forms for the loss function, similar to the MSE loss above. Many areas of physics suffer from this because these ansatze diverge in many cases or fit the data poorly. Hence, more complex L-conv loss functions can potentially serve as more advanced ansatze for solving physics problems."
>I would like you to describe what kind of physical problems you can solve here as well.
Personally, since the estimation of the Lie ring is important in the conservation law estimation of dynamical systems, I am interested in whether the method can be applied to such problems.
Also, if the authors use quantum field theory as the target of the discussion, I would like you to discuss whether discrete symmetry can also be treated in the framework of the proposed method.
That is because quantum field theory has many discrete symmetries.

>6-1 and 6-2 are related to the concerns in point 5.


There are some minor comments as described below:
>1. I understand that the Lie rings estimated by the method are restricted to those realized by linear transformations.
This restriction is looser than that of the representation of Lie groups.
Is there any advantage to this relaxation?

>2. Figure 4 is not cited in the text.

>3. (Line 197) Equation 58 is not found in the text.
Also, Equation 58 in Supplemental material does not seem to be the relevant equation.

>4. (Line 262) "assme" -> "assume"  typo?

>5. (Line 286) "here s" -> "here is"  typo?

>6. (Line 86) $\epsilon ^i$ seems to be undefined.


References:

[1] Mototake, Yoh-ichi. "Interpretable conservation law estimation by deriving the symmetries of dynamics from trained deep neural networks." Physical Review E 103.3 (2021): 033303.

[2] Zhou, Allan, Tom Knowles, and Chelsea Finn. "Meta-learning Symmetries by Reparameterization." International Conference on Learning Representations. 2020.

**Time Spent Reviewing:**

more than 12 hours

---

> ### Author Response · Authors · 2021-08-08
> **part 1**
>
> Thank you. We will do our best to clarify the purpose, which is to develop a universal building block and methodology for encoding and learning any continuous symmetry in neural networks.
>
> __We propose the Lie algebra convolutional network (L-conv), a building block for constructing group equivariant architectures.__
> > ... estimation error $O(\eta^{p+1})$...
>
> Thank you for sharing this very interesting paper. To clarify, $O(\eta^2)$ is the order of error for a single layer L-conv. Recurrant and multi-layer L-conv can approximate $G$ to arbitrary accuracy by approximating the $\exp$ map, given accurate $L_i$. The order of estimation error for an $n$ layer L-conv is $O(\eta^{n+1})$. One experiment on this is Appendix C, Fig. 6, reporting the accuracy of estimating finite translations using infinitesimal generators found using the Whittaker-Shannon interpolation. Figure 6 shows that with more recursion the error in L-conv becomes very small. The question about the curvature of the group manifold is interesting and merits a detailed study of non-abelian groups (recall that the curvature is $[L_i,L_j]$ and vanishes for Abelian groups such as $SO(2)$ and translations). But the error estimation remains the same: it's the error of expressing the exponential map as a Taylor series.
> When estimating a group element $g=P\exp[\theta^iL_i]$ with an $n$ layer L-conv the error is $O(\eta^{n+1})$ where $\eta=\max{[\theta^i]}/n$ (because $\theta$ has to be broken into $n$ steps).
> When $L_i$ are unknown and being learned, the experiment in Fig. 3 and others in Appendix C provide a clue about the estimation error.
> In Figs. 3 and 7 a recurrent L-conv learning finite $g\in G$ learns a good representation of $\hat{L}$. The accuracy of learning the vector field $\hat{L}$ drops for larger inputs (Fig. 8). But this can be reduced using eq. 17, where the rotation vector field $\hat{L}$ is decomposed into learning a 2x2 $L$ and a $d\times k$ lift $g_\mu$. Finally, we note that when $G$ is known and when a good approximate implementation of $G$ into the architecture exists (e.g. CNN) using that implementation may be preferable over using L-conv, as the L-conv achieving the same accuracy may be need to be very deep or recur many times. However, L-conv is most beneficial when the symmetry is unknown.
>
>
> __We show that multi-layer L-conv can approximate group convolutional layers, including CNN, and find graph convolutional networks to be a special case of L-conv.__
> > This is a very interesting discussion, but the meaning ... unclear.
>
> We agree about the relevance of the Appendix D results on higher performance on tasks with unknown symmetries, such as scrambled, rotated images. If you wish so, we will report some of those results in the main paper. We moved them to the end because they were preliminary and implementing the equivalent of an advanced model architecture, such as a ResNet, using L-conv requires more research. That is why the experiments in D are limited to one or two layers.
> Regarding the connection with GCN, the aggregation function in GCN treats all neighbors of a node the same way.
> L-conv, on the other hand, assigns different weights to each neighbor via $L_i$, similar to graph attention networks. For example, in images, the pixels are on a 2D lattice. If we use this lattice to make GCN, it only learns rotationally symmetric filters because it treats all neighboring pixels the same way. In L-conv, $L_i$ may be circulant matrices, one only receives messages from neighbors on the left of each pixel, one from upper neighbors, etc. Thus L-conv can extend GCN to include CNN. This can also be used to learn a CNN-like convolution pattern for unevenly sampled grids and point clouds. When the underlying space is a 2D mesh, the weighting of neighbors of each node is similar to Gauge CNN, as mentioned after eq. 17.
>
> __The Lie algebra generators in our L-conv can be learned to automatically discover symmetries.__
> > As far as I know, there are methods that also learn Lie groups [2] or estimate Lie-ring from trained DNNs [1]. What is the advantage of the proposed method?
>
> Zhou et al's Meta-learning Symmetries by Reparameterization (MSR) is certainly an interesting idea, but it has important differences with our method, specifically Proposition 1 in Zhou (2021).
> The main disadvantage of MSR, in our opinion, is the large number of parameters in $U^G$ to be learned.
> First, they work with a discrete group $G=\{g_1,\dots, g_n\}$.
> They decompose the weights $W\in \mathbb{R}^{s\times s}$ of a fully-connected layer, acting on $\mathbf{x} \in \mathbb{R}^s$ as $\mathrm{vec}(W) = U^Gv$ where $U^G\in \mathbb{R}^{s\times s}$ are the "symmetry matrices" and $v\in \mathbb{R}^s$ are the "filter weights".
> Then they use meta-learning to learn $U^G$ and during the main training keep $U^G$ fixed and only learn $v$.
> We may compare MSR to our approach by setting $d=s$ in the tensor notation (e.g. the number of pixels).
> First, note that although the dimensionality of $U\in \mathbb{R}^{nd\times d}$ seems similar to our $L \in \mathbb{R}^{n\times d\times d}$, the $L_i$ are $n$ matrices of shape $d\times d$, whereas $U$ has shape $(nd) \times d$ with many more parameters than $L_i$.
> Also, the weights of L-conv $W\in \mathbb{R}^{n\times m_{out} \times m_{in}}$, with $m_{out}$ being the number of channels, are much fewer than MSR filters $v\in \mathbb{R}^d$.
> Finally, the way in which $Uv$ acts on data is different from L-conv, as the dimensions reveal.
> The prohibitively high dimensionality of $U$ requires MSR to adopt a sparse-coding scheme, mainly Kronecker decomposition.
> Kronecker assumes a block structure for $U$, thereby heavily restricting the structure of symmetry generators.
> This block structure means only neighboring pixels mix, whereas in a problem such as scrambled-rotated images, the mixing is non-local.
> Though not necessary, we too choose to use a sparse format for $L_i$, finding that low-rank $L_i$ often perform well (note that for CNN, $L_i$ need to be circulant matrices, which are full-rank, but can be sparse-coded).
> Regarding learning Lie groups in trained DNN, we believe that is a different problem. The DNN would need to be general and expressive enough so that it learns to encode all the symmetries in the data (e.g. very wide MLP encoding all rotated versions of the same image). Our goal is to remove this requirement of large DNN, starting with a compact network which can learn and encode symmetries of the data with a fraction of the parameters of MLP (see FC results in appendix D).
>
>
> __We find that MSE loss function for L-conv generalizes important Lagrangians in physics.__
> > I could not understand why "The MSE is given by".
> The statement is specific to a single layer L-conv with a quadratic error. We do not claim that any MSE is related to symmetries, nor that only MSE is related to symmetries. Specifically, we are stating that the MSE for single layer L-conv is identical to one of the most commonly used physics Lagrangians, the free field theory Lagrangian. This Lagrangian has historical origins, generalizing quantum mechanics (QM) to quantum field theory (QFT). But this Lagrangian is also derived in a mathematically rigorous way from a Lie algebra in "sigma models" to create Lagrangians preserving the corresponding symmetry group.  That being said, the physical Lagrangians have an interpretation similar to likelihood maximization interpretation of MSE. In physics, the classical trajectory is the maximum likelihood trajectory and the minimizer of the action ( integral of Lagrangian). The likelihood of each trajectory is similar to a Boltzmann factor $\exp[i/\hbar \int \mathcal{L}]$, with Lagrangian $\mathcal{L}$.
>
> __We find generalization error and equivariance in the loss landscape can be expressed as Euler-Lagrange equations and Noether currents, transferring ideas from physics to ML.__
> > About the discussion about this novelty (from Line 283 to 296), it seems to be obvious ...
>
> Respectfully, we believe this point is actually much deeper. Let us discuss the Robustness using Euler-Lagrange (EL) and the Noether’s current part separately.
>
> __Euler-Lagrange:__
> There is a key distinction between how ML uses cost functions and how physicists are taught to use Lagrangians (equivalent of cost function). In ML, we use the data $\phi$ to fit the parameters in the cost function. In physics, we assume the parameters are given and find what $\phi$ would minimize the integral of the Lagrangian (EL equations). You see that these are two different optimizations: ML is optimizing the model parameters, physics is optimizing the _input_ $\phi$. Now we are suggesting that the EL equations can also be used in ML to assess robustness of the model. Specifically, random perturbations to a given input $\phi_n$ should keep the ML loss minimized. But adversarial attacks may violate the EL equations and this could prove useful for anomaly detection. To the best of our knowledge, no prior work has suggested using EL equations for assessing robustness of ML systems to perturbations and this could potentially become a new area of research.

---

> ### Author Response · Authors · 2021-08-08
> **part 2**
>
> __Noether’s currents:__
> Noether’s theorem is a powerful tool in physics for discovering symmetries. It states that whenever a system exhibits continuous symmetries, certain quantities are preserved during dynamics. This idea can be extended beyond dynamics to identifying flows which are divergence-free along the direction of symmetry. One big question for any physicist learning about the equivariance in ML literature is whether such conserved Noether’s currents also exist in ML, a question partly answered in the Kunin (2020) paper for general DNN. We found another interesting relation: the connection with field theory gave a direct way to derive Noether’s currents for equivariant networks, meaning we can say precisely what quantity derived from the loss function is preserved under symmetry transformations. Thus, Noether’s theorem could be used to introduce a new way to find symmetries. Right now in L-conv to find symmetry generators $L_i$ we perform SGD on the full model and hope that it identifies the correct $L_i$. Using Noether’s theorem, we would have an additional equation  (eq. 23 or eq. 60) that could be minimized to find $L_i$, perhaps using meta-learning.
> Lastly, in the extended derivation in Appendix A.4 we show that the conserved quantity in L-conv with MSE cost is what physicists call the “stress-energy” tensor. In physics this tensor is always conserved for spatial (or space-time) symmetries. An immediate follow-up to this is, what conserved quantities do we find if the symmetries are not spatial (e.g. generalizing steerable and gauge CNN). These currents can reveal the flow of symmetry directions in the loss landscape and identify valleys of low loss.
>
> > There are the other major comments about "7 Discussion"
>
>
> __6-1. Response:__
>  As we mentioned in the previous answer, physicists have used this L-conv MSE loss in a very different way that ML. One advance in ML could be the introduction of Eeuler-Lagrange equations for assessing robustness, detecting adversarial attacks, and even as a generative model, as the $\phi$ satisfying EL are data points fitting the probability distribution of the dataset.
> Another direction is using Noether’s theorem to learn symmetry generators, identify conserved quantities. These can help reduce the dimensionality of the low-loss landscape and provide new, symmetry-aware dimensionality reduction methods. Knowledge of the symmetry structure of the low-loss landscape could also improve how GAN are trained and work, as they could be forced to learn only the non-symmetry directions by moving orthogonal to the Noether current.
>
> __6-2. Response__
> Specifically, we believe modelling strongly interacting systems, turbulence and dark matter vs modified gravity are examples of models where the traditional physics model building has failed. Modeling dynamical systems too could benefit from symmetry discovery. For instance, finding invariant tori for systems beyond simple harmonic motion has been a challenge, but L-conv may be able to learn the Hamiltonian flow in such systems.
> In general, the process of modeling field theories in physics usually involves adding polynomial terms to the free field Lagrangian, which is MSE of L-conv, or combining primitive functions. But in ML, model building involves connecting multiple layers in sophisticated ways which can yield functions not easily expressible as polynomials. It is well possible that many physical phenomena cannot be described by adding more terms to free field theories. For example, protons have the same $SU(3)$ symmetry as quantum Chromodynamics (QCD), a weakly interacting field theory describing quarks and gluons. However QCD cannot model protons because the interactions inside protons is strong.
> In fact, the best models for nucleon interactions are phenomenological, fitting a few numbers to learn how they wobble during interactions. Using ML and L-conv, one could make a black-box model which is guaranteed to have the $SU(3)$ symmetry, but which is not a polynomial extension of QCD (e.g. multi-layer L-conv, combined with other DNN). It also does not need to have MSE loss.
>
> >  discrete symmetry ... because quantum field theory has many discrete symmetries.
>
> Are you referring to the charge conjugation, parity and time-reversal (CPT) symmetries? These are elements of the Lorentz group and map the group manifold component containing the identity to other components.  Such discrete symmetries don't have a corresponding $L_i$, but can be encoded using $gL_i$ where $g$ is each of the discrete symmetries. We can work out more details of this if you wish. But if you are referring to other symmetries, please clarify this point.
>
> >There are some minor comments as described below:
> I understand that the Lie rings ... linear transformations...this relaxation?
>
> Actually group representations are defined as a map to linear transformations. Note that this includes the adjoint representation $gAg^{-1}$, and as functions can be thought of as infinite dimensional vectors, infinite dimensional group representations can exist too. If you question is something else, can you please clarify?

---

### Author Response · Authors · 2021-08-09
**General note**

We would like to thank all the reviewers for their insightful comments. We are glad to see that the majority of reviewers find our work important. We appreciate that  Reviewer 2v7e believes that "the method is very novel and significant in the study of equivariant neural network", Reviewer DTit says it "presents very interesting ideas and has high theoretical value", and Reviewer vpfc states "I can assure that the mathematical instrumentation of their method is solid".  We are grateful to hear Reviewer 2v7e believes that if their comments are addressed the paper has "both significant novelty and importance that should be urgently shared with researchers" and Reviewer DTit states "the vastness of sound theoretical insights makes the paper in my opinion worthwhile a publication at NeurIPS"

Nevertheless, all reviewers also point out important comments stating that:
1) the presentation of key results (e.g. eq. 6) needs improvement,
2) the paper is densely packed with technical discussion, making it difficult to read,
3) implementation details (eq. 16 and appendices C and D) should be elaborated and included in main paper,
4) Insights from section 6 need more explanation.

Given the novelty of the approach, technical details were necessary to lay the foundation and to ensure the rigorousness, at the expense of readability. At the same time, we did not want to leave out important insights gained from this approach (e.g. sec. 6), hence the density. Below we will address the individual points raised by the Reviewers in detail.

---

### Decision · Program_Chairs · 2021-09-27

**Decision:**

Accept (Poster)

**Comment:**

In this work, the authors build group equivariant neural network layers. These are built from convolutions with sufficiently local functions that can be lifted from the Lie group to the Lie algebra. This approach is sufficiently general to encompass previous equivariant architectures. This approach also comes with a universal approximation property, which the authors need to highlight.

== Why accept this paper?
The paper offers a clearly creative and novel contribution, with a number of high value theoretical hindsights. The community will greatly benefit from having access to these ideas, and I look forward to seeing what and how the present authors, or other researchers, will build on top of those ideas.

== Why not a higher rating?
As mentioned by all reviewers, this paper is particularly hard to read. Part of the reason is that it deals with a difficult problem, but we can only praise the authors for this, not blame them. But another reason for the difficulty to read the paper is its very high density with a lack of focus and clarity. Following the discussion with reviewers, the authors received a number of suggestions to improve their paper, and they should implement as many of those suggestions as possible.